# VLMgineer: Vision Language Models as Robotic Toolsmiths

**George J. Gao**[†], **Tianyu Li**[†], **Junyao Shi, Yihan Li**[‡], **Zizhe Zhang**[‡],
**Nadia Figueroa, Dinesh Jayaraman**
GRASP Lab, University of Pennsylvania

## Abstract

Tool design and use reflect the ability to understand and manipulate the physical world through creativity, planning, and foresight. As such, it is often regarded as a measurable indicator of cognitive intelligence across biological species. While much of today's research on robotics intelligence focuses on generating better control strategies, inventing smarter tools offers a complementary form of physical intelligence: moving the problem-solving onus into the tool's geometry so that control becomes simpler. This motivates us to ask: can today's foundation models offer useful priors to automatically invent—and effectively wield—such tools? We present VLMgineer, the first fully automatic framework designs tools and actions from scratch by harnessing the creativity of Vision–Language Models (VLMs) together with evolutionary search. We evaluate VLMgineer on a diverse benchmark of everyday manipulation scenarios that demand creative tool design and use. Across this suite, VLMgineer consistently discovers tools and policies that solve tasks more effectively and innovatively, transforming challenging robotics problems into straightforward executions. It also consistently outperforms VLM-generated designs from human specifications and existing human-crafted tools for everyday tasks. We further demonstrate that VLMgineer's automatically designed tools and action policies transfer seamlessly to real-world task execution on a physical robot. To facilitate future research on automated tool invention, we will release our benchmark and code. Project Website: vlmgineer.github.io.

## 1 Introduction

Humans exhibit a remarkable ability to design and utilize tools, fundamentally extending their capabilities to accomplish tasks otherwise beyond their reach through creativity, planning, and foresight. This capacity for tool creation and usage represents one of our most distinctive cognitive adaptations, and therefore is widely regarded as a marker of cognitive complexity. Achieving comparable versatility in robots demands a coupled approach: the shape of a tool and the motions that wield it should be co-designed — each constraining and enabling the other. Much of today's robotics research concentrates on enabling complex robot motions that use simple standard tools (Shi et al., 2023; Qi et al., 2024; Car et al., 2024; Shaw et al., 2024; Chen et al., 2024). In this work, we pursue an alternative form of physical intelligence: inventing *smarter* tools that simplify downstream control — thereby shifting the primary problem-solving burden from devising control strategies to designing the tool's geometry.

State-of-the-art vision–language models (VLMs) possess vast and impressive common-sense reasoning and creative abilities, alongside extraordinary capabilities in code generation, visual comprehension, and in-context learning. When combined with evolutionary search methods, VLMs have successfully crafted human-level reward functions for reinforcement learning (Yu et al., 2023; Ma et al., 2023), 3D graphics (Huang et al., 2024), articulations of in-the-wild objects (Le et al., 2024), intricate 3D sculptural designs (Goldberg et al., 2024), and developing advanced algorithms to solve mathematics and science problems (Romera-Paredes et al., 2024; Aglietti et al., 2024; Novikov et al., 2025).

---

† and ‡ denote equal contribution. Correspondence: {gegao, tianyu}@seas.upenn.edu

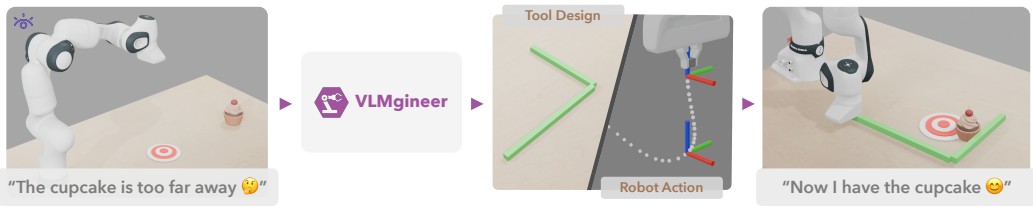

Figure 1: Given a manipulation task that lies outside the robot's capabilities, **VLMGINEER** first prompts a vision language model to generate a tool and action. We then employ evolutionary search in simulation to refine the tool's geometry and synthesize the corresponding robot motion plan. Finally, the robot, equipped with the automatically designed tool, successfully completes the task.

In the wake of these results, we ask: *can today's VLMs also guide the design of innovative and action-efficient physical tools for robots*? We introduce VLMGINEER, an autonomous framework that leverages VLMs to jointly evolve both tool design and manipulation strategies for robots. Our method demonstrates unprecedented efficacy in developing specialized tools for diverse manipulation tasks, through an evolutionary search process guided by VLM-generated tool geometries and action plans. VLMGINEER is the first fully automatic framework for designing tools and actions from scratch: compared to prior more limited investigations of tool design, that largely consider parameter optimization for a manually designed parametric template, VLMGINEER works off-the-shelf for new tasks without human-in-the-loop steps such as task-specific templates, prompts, or examples. To facilitate future research and benchmarking, we also introduce ROBOTOOLBENCH, a comprehensive simulation suite comprising 12 diverse robotic tool-use manipulation tasks specifically designed to evaluate tool design and policy optimization methods.

In summary, we make the following contributions:

- **VLMGINEER, a novel evolutionary optimization framework** that automatically discovers innovative tools to solve robotics task more efficiently.
- **ROBOTOOLBENCH, a comprehensive simulation benchmark** consisting of 12 robotic tool-use tasks designed explicitly for evaluating robotic tool and policy designs.

**Our fully autonomous approach not only outperforms designs generated with human specifications and human-crafted everyday tools, but also produces tools and actions in simulation that seamlessly transfer to real-world task execution.** When evaluated on ROBOTOOLBENCH, VLMGINEER achieves an average normalized improvement of 64.7% over VLM-generated designs from human language specifications and outperforms existing human-crafted tools by an average normalized improvement of 24.3%. Our results serve to validate both the physical design intelligence enshrined in VLMs pre-trained on web-scale data, demonstrate amplification of VLM physical creativity via evolutionary search beyond prompting, and present the promise of more adaptable and capable robotics systems that can ingeniously create and use tools.

## 2 RELATED WORK

**Task-specific computational agent and tool design.** Previous research has extensively investigated methods for optimizing robot morphology, end-effectors, and tool designs for robot manipulation through various computational approaches, ranging from model-based optimization (Allen et al., 2022), reinforcement learning (RL) (Li et al., 2021), data-driven generative models (Wu et al., 2019; Ha et al., 2020; Xu et al., 2024), and differentiable simulation (Li et al., 2023). Others have explored robot design for locomotion using evolutionary algorithms (Jelisavcic et al., 2019; Hejna III et al., 2021; Walker & Hauser, 2021; Sims, 2023; Dong et al., 2023a;b), stochastic optimization (Exarchos et al., 2022), and graph search (Zhao et al., 2020). However, these existing approaches typically require manual task-specific pre-definition of a handful of optimization parameters, rely on fixed trajectories or pre-defined control policies, and tend to suffer from low sample efficiency. Prior efforts like Khan et al. (2025) have demonstrated that VLMs can generate functional tool designs, often resembling existing objects found in online databases. In contrast, our core novelty lies in the finding that evolutionary search can elicit physical creativity from VLMs significantly beyond

single-shot prompting. This process allows our system to discover nonstandard, highly performant geometries that are not merely retrieved but iteratively optimized for specific tasks. As demonstrated quantitatively in Fig. 7 and Table 1, our evolutionary approach yields tools that are significantly more effective than those from the initial VLM-generated population, confirming that the iterative refinement is crucial for grounding the design in physical performance. We also introduce a VLM-driven approach that simultaneously optimizes both tool design and manipulation policies, enabling generalization across diverse manipulation tasks without requiring manual parameter specifications.

**Robot learning for tool-based tasks.** To learn effective tool usage, some have employed learned or simulated dynamics models for tool manipulation optimization (Xie et al., 2019; Allen et al., 2020; Girdhar et al., 2020; Lin et al., 2022a;b). Another prevalent approach involves learning tool and object affordances — understanding the functions of objects and tool-object interactions (Fang et al., 2020; Qin et al., 2020; Brawer et al., 2020; Xu et al., 2021a; Noguchi et al., 2021; Shi et al., 2023). Recently, large language models have been leveraged to employ creative tool use (Xu et al., 2023). While such methods typically assume that suitable tools already exist in the environment, we instead address the more practical scenario where a general-purpose robot must concurrently optimize both the tool's design and its manipulation strategies.

**Joint optimization of morphology and control.** Jointly addressing tool design and control problems has often involved formulating nonlinear programs to solve task and motion planning (TAMP) given predefined design parameter space, which are particularly effective for sequential manipulation over extended horizons (Toussaint et al., 2018; 2021). However, given our objective to deploy VLMGINEER in any arbitrary environment without manual specification of design parameters, we rely on the underestimated physical creativity of VLMs. Approaches using RL (Wang et al., 2023a;b; Luck et al., 2020; Yuan et al., 2021), gradient-based optimization (Spielberg et al., 2019), Bayesian optimization, evolutionary algorithms (Cheney et al., 2018; Mertan & Cheney, 2024; Ringel et al., 2025), or a combination of them (Liao et al., 2019; Schaff et al., 2019; Ha, 2019; Bhatia et al., 2021; Pathak et al., 2019) have been proposed for joint morphology and control learning for robot locomotion tasks, in particular with soft or modular robots. Studies on joint robot or tool and policy design through RL (Chen et al., 2020; Liu et al., 2023), differentiable simulation (Xu et al., 2021b), and model-based optimization (Kawaharazuka et al., 2020) have also demonstrated effectiveness in tool manipulation. However, since these methods still all require manual specifications of the design space, they require significant human efforts to scale beyond a few tasks.

Recent work has explored LLM-aided evolutionary search for robot design in conjunction with RL-based policy optimization in locomotion (Qiu et al., 2024; Song et al., 2025), demonstrating the potential of using LLMs to unlock more performant robot design. Unlike prior work, our work targets open-world VLM-guided design of both tools and actions for manipulation without human-in-the-loop parameter specification. **VLMGINEER leverages the surprising physical creativity of VLMs to automatically create design solutions using evolutionary search.** It can easily be scaled to a wide range of tasks, and it is much more efficient in terms of samples, time, and compute than prior RL-based methods.

## 3 BACKGROUND

**Evolutionary Methods.** Evolutionary algorithms (Langdon & Poli, 2013; Doncieux et al., 2015) have a long-standing history in solving optimization problems, inspired by principles of biological evolution and natural selection. They are particularly effective in black-box optimization with vast optimization spaces, such as open-ended design. At their core, these methods maintain a **population** of candidate solutions, which iteratively evolve through carefully designed mutation and crossover operators. Each iteration evaluates individuals against a **fitness function**, selecting those with higher fitness while discarding or replacing less successful candidates. To balance exploitation and exploration, **crossover** combines promising solutions into offspring, and **mutation** introduces novel variations. Evolutionary algorithms have proven effective across diverse domains such as program synthesis, symbolic regression, algorithm discovery, and even robot design. Nevertheless, their reliance on handcrafted mutation and crossover operators remains a significant limitation—such operators are challenging to design and often inadequately capture essential domain-specific insights.

**Large model-guided evolution.** To improve the scalability, performance, and automation of evolutionary algorithms, recent work has integrated large models into the evolutionary process, automating

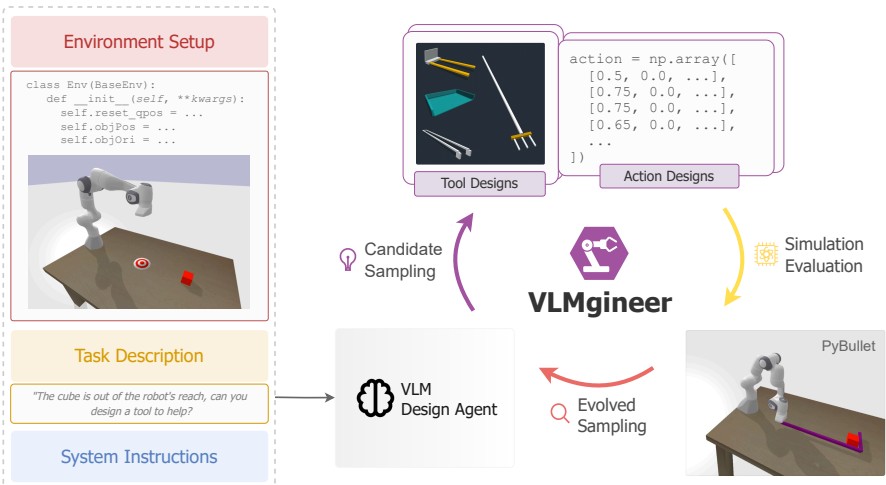

Figure 2: VLMGINEER takes unmodified environment source code, environment image, environmental description, and task description as context to zero-shot generate tool and action designs from a VLM. It then iteratively refines its tool and action designs through a loop of candidate sampling, simulation-based evaluation, and evolution improvement.

mutation and crossover operations. Leveraging the extensive world knowledge and inductive biases inherent in large models allows for more efficient evolution of candidate solutions and also eliminates the necessity of manually defining allowed mutation operations. Moreover, some approaches exploit the rich semantic understanding of large models to provide nuanced, semantic feedback beyond simple numerical fitness scores. Specific implementations of these principles in evolutionary algorithms vary according to the domain. For instance, Ma et al. (2023) employs large language models (LLMs) to guide evolutionary reward design in reinforcement learning. Eureka generates a *population* of candidate reward functions from raw environment code, evaluates RL agents trained with these rewards using a task-specific *fitness* function, and selects the best-performing candidates. It then employs LLM-guided in-context reward *mutation*, proposing improved functions based on textual feedback. Drawing inspiration from this, we investigate whether vision–language models (VLMs) can similarly provide inductive biases to guide the evolutionary design of robotic tools and actions.

## 4 METHOD

VLMGINEER builds upon previous Large Model-guided evolution methodologies to perform tool-action co-design. Specifically, VLMGINEER consists of three algorithmic components: (1) We prompt the VLM to generate a diverse **population** of potential candidate tool-action samples given raw environment code, task description, and system instructions as context. (2) We evaluate each of the design samples via task **fitness** functions and retain those with the top-$k$ rewards. (3) We iteratively prompt the VLM to produce novel tool-sample offspring via guided tool **mutation** and **crossover**, progressively improving tool and action designs. This overall VLMGINEER algorithm is summarized in Algorithm 1. Please refer to Appendix A.4 for the full prompts.

**Joint tool and action candidate sampling.** While previous approaches of large model-guided evolutionary robot design (Qiu et al., 2024; Song et al., 2025) typically optimize robot

---

**Algorithm 1** VLMGINEER: Evolutionary Tool and Action Co-Design with VLMs

---

**Require:** Environment code $\mathcal{E}$, image render $I$, task description $d_{\text{task}}$, fitness function $\mathcal{F}$, initial prompt PROMPT, Vision-Language Model VLM
1: **Hyperparameters:** Number of evolution cycles $n$, population size $K$, top-$k$ selection threshold
2: **for** $n$ iterations do **do**
3:     // **Sample K designs**
4:     $D_1, D_2, ..., D_K \sim \text{VLM}(\mathcal{E}, I, d_{\text{task}}, \text{PROMPT})$
5:     // **Evaluate design candidates**
6:     $s_1 = \mathcal{F}(D_1), \cdots, s_K = \mathcal{F}(D_K)$
7:     // **Selection**
8:     Select top-$k$ designs $\{D_{j_1}, ..., D_{j_k}\}$ with highest $s_j$
9:     // **Evolution**
10:     PROMPT := PROMPT : EVOLUTION PROMPT$(\{D_{j_1}, ..., D_{j_k}\})$
11: **end for**
12: **return** Final design $D^* = \arg\max_D \mathcal{F}(D)$ across all iterations

---

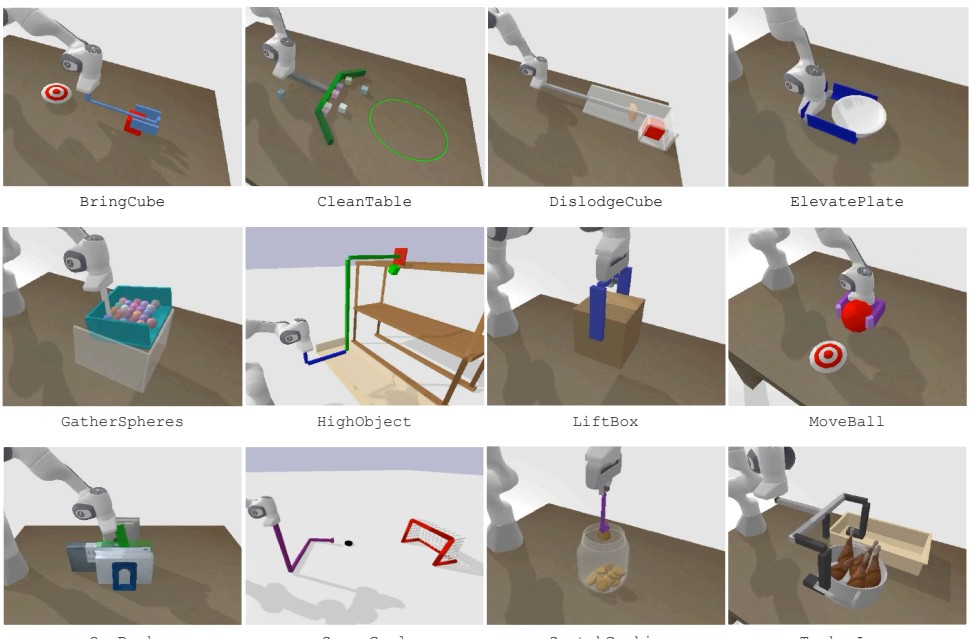

Figure 3: VLMGINEER produces innovative tool designs and their corresponding actions across 12 diverse tasks in ROBOTOOLBENCH that are challenging to perform using a general-purpose robot arm and gripper.

morphology alone, relegating action or control optimization to a subsequent evaluation stage. Our approach prompts the VLM to simultaneously generate paired tool designs and corresponding action strategies in a single inference step. Our key insight behind joint tool-action sampling is that it allows for a tighter coupling between tools and their associated actions. Rather than sequentially optimizing the tool geometry first and then actions afterward, simultaneous optimization leverages the VLM's inductive biases to smoothly navigate the joint tool–action design space towards the Pareto frontier. Concretely, within each evolution cycle, VLMGINEER prompts the VLM to propose $n$ distinct tool designs along with $m$ candidate action plans per tool, resulting in $n \times m$ total tool-action pairs. This corresponds to a kind of crude VLM-guided policy optimization, which merely selects the best among the $m$ generated action plans. Compared to policy optimization via RL (Ma et al., 2023; Song et al., 2025; Qiu et al., 2024), our action sampling approach, albeit simple, significantly accelerates iteration cycles and reduces computational overhead by exploiting the insight that appropriately designed tools inherently simplify and enhance action plans. We also empirically observed this strategy to outperform other forms of feedback, such as videos or object-centric signals, whose low accuracies tend to degrade performance and limit meaningful improvement.

**Specification of crossover and mutation.** A critical part of how VLMGINEER enables effective tool design evolution is the utilization of *inductive in-context **crossover and mutation***. We define inductive in-context crossover and mutation as the process of prompting VLMs to introduce random, free-form tool mutations and crossovers, conditioned on previous elite tool candidates, and guided by the model's learned inductive biases for producing better task-solving tools. We use the prompt below to perform inductive in-context crossover and mutation: *"Your design decision is part of a genetic algorithm for tool creation, where each new design is produced either by mutation—changing exactly one aspect (e.g., adjusting a component's dimension or adding/removing a component)—or by crossover, combining elements from two existing designs. All resulting mutations and crossovers should plausibly enhance task success while preserving design diversity."*

**Tool representation format.** Selecting an appropriate representation for tools—balancing abstraction, design flexibility, and manufacturability—is critical for effective optimization. Prior works have represented objects and tools as meshes (Nair et al., 2020), CAD (Thomas et al., 2018), or blocks (Goldberg et al., 2024). These representations, however, either introduce excessive complexity and optimization challenges or lack sufficient expressiveness. Inspired by prior work (Le et al., 2024), we represent tools in Unified Robot Description Format (URDF). The structured, modular nature of

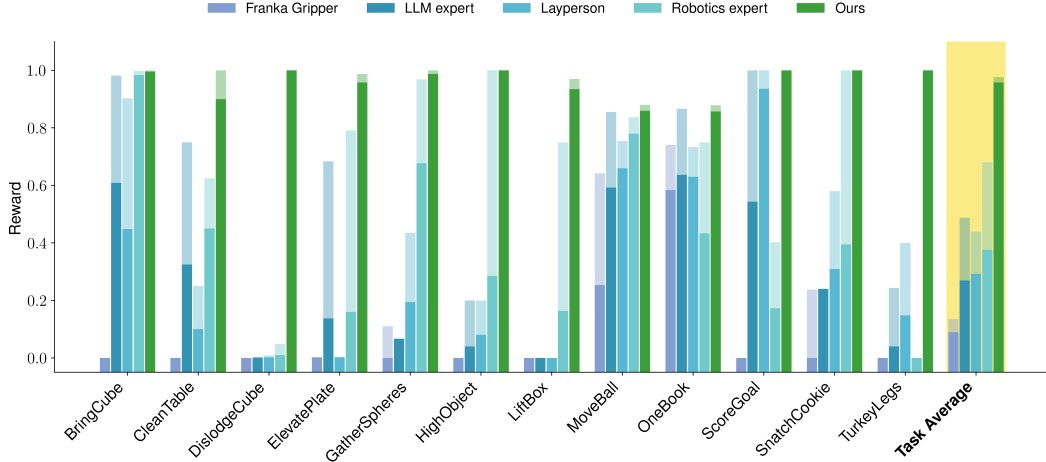

Figure 4: Comparison of rewards for the Franka Gripper, 3 Human Prompt experiments, and our proposed method across 12 tasks. Darker bars indicate the *average* reward over five runs, while paler bars indicate the *best* reward. All results here were using `gemini-2.5-pro-preview-03-25`.

URDF, analogous to code blocks, aligns seamlessly with vision–language models' (VLMs) strengths in code understanding and generation. Concretely, we prompt the VLM to generate URDF-defined tool designs as modular blocks that can be directly integrated into a designated end-effector link of the robot model.

**Action representation format.** Building on recent work that leverages VLMs for action generation (Di Palo & Johns, 2024; Yin et al., 2025), we prompt the model to explicitly output action sequences in the form of an $N \times 7$ array, where $N$ denotes the number of waypoints. Each row encodes a 6-DoF pose for the robot end-effector, along with a gripper open/close command. In this work, we intentionally use discrete waypoints to show that smarter tools could reduce the need for sophisticated policies. When needed, however, our framework is feasible for dynamic or force-based actions (e.g. force vector or wrench estimate for force-based reasoning) and can warm-start expressive closed-loop policies.

## 5   ROBOT TOOL DESIGN BENCHMARK

We propose a comprehensive simulation benchmark **ROBOTOOLBENCH** designed explicitly for evaluating robotic tool and policy design. ROBOTOOLBENCH comprises 12 object manipulation tasks designed to be challenging for the conventional robot morphology to complete. These task environments are visualized in Fig. 3. For several tasks (`BringCube`, `CleanTable`, `GatherSpheres`, `ScoreGoal`), we took inspiration from the subset of RLBench (James et al., 2020) tasks that involve tool use — note, however, that we expect that automated tool design will *replace and improve* the original tools from RLBench. Several other tasks (`HighObject`, `ElevatePlate`) are inspired by prior works in computational co-design (Liu et al., 2023) that study task-specific design parameter optimization as discussed in Sec 2. Still more task environments are inspired by everyday home scenarios (`LiftBox`, `MoveBall`, `OneBook`, `SnatchCookie`, `TurkeyLegs`). Finally, `DislodgeCube` is inspired by a tool design behavior previously observed in the Caledonian crow (Jacobs et al., 2016), which used tools to retrieve objects in confined spaces. We adopt the Franka Panda robot arm as the standard morphology to attach tools to, and implement our environments using PyBullet (Ellenberger, 2018–2019). For more details of the each task, please refer to Appendix A.2.

## 6   EVALUATION

Our experiments are designed to provide a comprehensive analysis of VLMGINEER's capabilities. We aim to answer the following questions:

- **Q1:** Can VLMGINEER effectively discover innovative tools and the actions to use them across a diverse set of manipulation tasks?
- **Q2:** How does VLMGINEER's autonomous co-design compare to tool designs specified to a VLM by human users with different expertise?
- **Q3:** How important is the evolutionary framework to VLMGINEER's performance?
- **Q4:** Can the co-designed tools and actions be successfully transferred from simulation to solve tasks in the real world?

## 6.1 PERFORMANCE ON ROBOTOOLBENCH

**Baselines.** To showcase VLMGINEER's ability to generate creative and effective tools and usage actions, we compare our method with the following baselines: **(1) Franka Gripper**: We evaluate the performance of the vanilla Franka Panda two-finger gripper without additional tools on ROBOTOOL-BENCH to highlight the inherent limitations of the robot's default morphology; these tasks are after all explicitly designed to be very hard or impossible to perform without the right tools. We derive the no-tool action policy by prompting the VLM to follow an action-sampling procedure analogous to our proposed method, minus the use of any tools. **(2) Human Prompts**: For these baselines, we ask humans to specify a tool design to the VLM in natural language, following which it attempts to generate that tool and several action plans, as in our method. There is no evolutionary search. We evaluate on humans with varying expertise: "Robotics expert" (a graduate student researching robot learning), "LLM expert" (a graduate student researching LLMs), and "Layperson" (an undergraduate student with no relevant research experience). **(3) RLBench Tools**: We evaluate four original tools from the tasks we adapted from RLBench, which are often natural everyday tools for the tasks considered. While RLBench tools are existing tools for everyday tasks, they might not have been explicitly optimized for maximum task success. However, as illustrated in Fig. 5, they represent sensible and common designs used in practice, making them practical and meaningful benchmarks. For further details of our experimental setup, see Appendix A.1. For consistency, all evaluations were done using `gemini-2.5-pro-preview-03-25`.

Note that a key distinction between VLMGINEER and other studies in the Related Work section is that the other studies all involve substantial manual parameter tuning or predefined parametric tool designs, which are fundamentally different from VLMGINEER's fully automated approach. Hence, direct apples-to-apples comparisons would be challenging. In fact, VLMGINEER could serve as a strong prior for these related works.

**Evaluation Metrics.** To assess the quality of a tool-action design after each execution, we define the evaluation metric as Task Rewards, which is a set of pre-defined task reward functions $R : S \to r \in [0, 1]$ that are unique to each task, where $S$ is its environmental state and $r$ is a normalized reward. These rewards are designed to evaluate the progress made in the task by a certain tool-action pair.

**Results.** The results are summarized in Fig. 4. VLMGINEER works consistently well across tasks, in terms of both average and best rewards. We dive into interesting individual method comparisons now. As expected, the default Franka Panda two-finger gripper fails on the majority of these tasks. What is perhaps more noteworthy is that **VLMGINEER outperforms human-prompting.** This is true even for expert humans across all tasks and on both metrics (better peak performance and also more reliable). While human prompts occasionally produced strong solutions, their results were less consistent and efficient. In tasks like `CleanTable` and `ScoreGoal`, both approaches reached similar peak rewards, but our method did so with significantly shorter paths. For further analysis, Fig. 5 shows example designs from human-prompting and VLMGINEER. Human-designed tools (left column) generally offer suitable forms for task completion; however, VLMGINEER (right column) creates more specialized features that enhance performance. For instance, in task `ScoreGoal`, our method produces long and bent shapes facilitating simpler, more efficient motions, which the robot just need to move very little along one axis to hit the puck. On the other hand, the straight tool designed from human prompts would require more careful control of the puck. In `GatherSphere`, our design includes a scoop with side protection and an overhead stripe structure, effectively preventing spheres from bouncing away.

**VLMGINEER tools also outperform the RLBench original tools**. On the four RLBench-based tasks, we evaluated the standard RLBench Tools (Fig. 5 middle column). As shown in Fig. 6, across

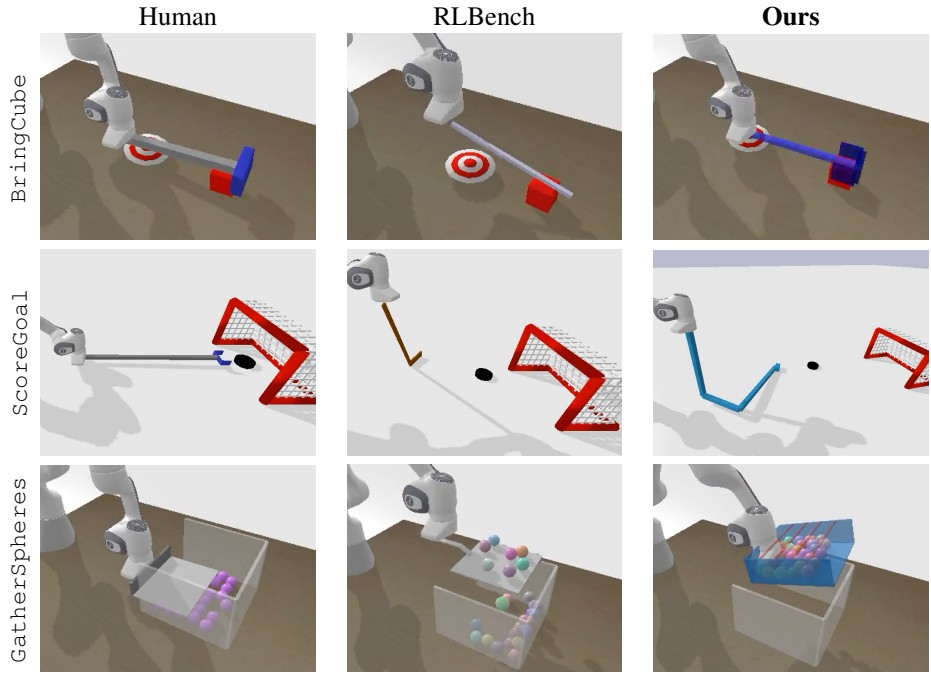

Figure 5: Qualitative comparison of human-designed, RLBench, and VLMGINEER tools on three tasks: `BringCube` (top row), `ScoreGoal` (middle row), and `GatherSpheres` (bottom row).

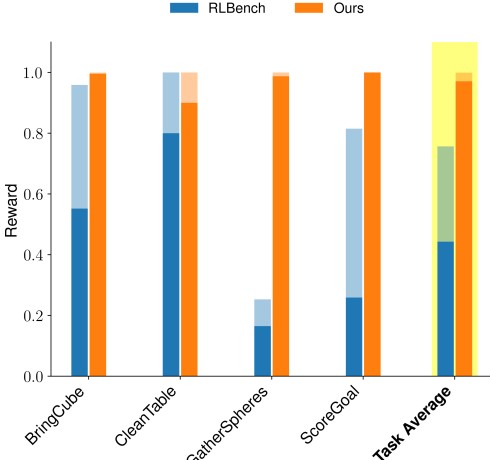

Figure 6: Performance comparison against standard RLBench tools. VLMGINEER consistently outperforms the original, human-crafted tools across four relevant tasks.

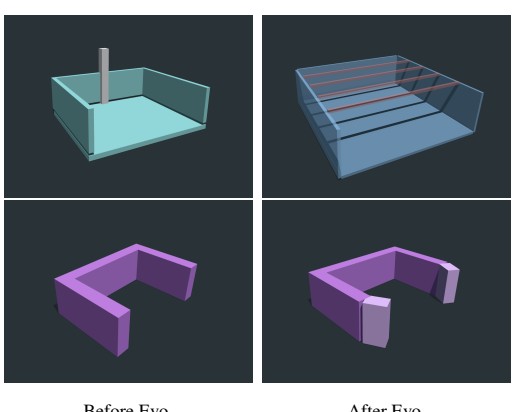

Figure 7: Qualitative examples of evolutionary refinement. The process makes intuitive and effective improvements to initial tool designs for tasks like `GatherSpheres` (top) and `MoveBall` (bottom).

every task, VLMGINEER not only attains the highest possible reward but does so more reliably (on average) than RLBench Tools. Qualitatively inspecting the tools further highlights the advantages of our method. The RLBench tools, originally designed for similar but distinct tasks, often underperform due to less optimized features. For example, in `BringCube`, the RLBench's simple stick provides insufficient lateral control, resulting in inconsistent cube manipulation. Our method's cage-like structure reliably locks and moves the cube closer, achieving significantly higher rewards.

## 6.2 Evaluating The Role of Evolutionary Search

To isolate the contribution of our evolutionary framework, we conducted an ablation study. Across three trials for each of the three chosen RoboToolBench tasks, we tested VLMgineer's standard evolutionary process (4 iterations of 2000 samples) against a brute-force baseline that used VLM sampling for 8000 evaluations, ensuring an identical sample budget for both methods. The results, presented in Table 1, show a clear advantage for the evolutionary approach. On average, **the evolutionary search strategy outperformed the sampling baseline by a significant 119.2%**

This quantitative advantage can be understood through the qualitative nature of the evolutionary process. We consistently observed evolution making intuitive and effective enhancements to initial designs. As illustrated in Fig. 7, for the `GatherSpheres` task, an open scoop was refined with guardrails to prevent spillage. Similarly, for `MoveBall`, an open-ended pusher was augmented with a hugging rim to improve control. This suggests that iterative refinement is a key mechanism that allows VLMgineer to discover robust and high-performing tool designs that are difficult to find through simple sampling.

Table 1: Mean normalized reward (0–1) for Evolution vs. VLM Sampling under an Equal Sample Budget. The structured search of VLMgineer substantially outperforms the brute-force sampling baseline, highlighting the critical role of the evolutionary framework.

| Method | ElevatePlate | RetrieveHigh | CleanTable | Average |
|--------|-------------|--------------|------------|---------|
| VLMgineer (Evolution) | **0.925** ± 0.007 | **1.000** ± 0.000 | **0.888** ± 0.018 | **0.938** (+119.2%) |
| VLM Sampling (Baseline) | 0.317 ± 0.254 | 0.501 ± 0.705 | 0.466 ± 0.124 | 0.428 |

## 6.3 Sim-to-Real Transfer and Real-World Validation

To validate that our VLMgineer generated design and actions work in real robotic scenarios, we selected three tasks—`MoveBall`, `ElevatePlate`, and `GatherSpheres`—to transfer zero-shot to a Franka robot. After running VLMgineer in simulation, we replicated the simulated environment in the real world, 3D printed the best manufacturable tool, mounted the tool on the robot, and played the associated action plan. The only minor modification we apply is to trim any portion that overlaps with the tool mounting head, a standard post-processing step. We present snapshots of our real-world experiment execution in Figure 8. More details of our sim-to-real transfer process and real can be found in Appendix A.9. In our real-robot experiments, for each task, we recorded the normalized rewards by executing the co-design for 5 runs to account for any environmental variance during execution, and obtained an average normalized reward for each task's winning co-design. The average normalized rewards from sim-to-real transfer experiments for `MoveBall`, `ElevatePlate`, and `GatherSpheres` are 0.959, 0.761, and 0.713, respectively. Overall, we observe VLMgineer's tools and actions effectively translate to real-world scenarios. Please refer to our website for the real-world videos.

## 7 Conclusion

We propose VLMgineer, the first fully autonomous framework for co-optimizing tool design and tool use actions by leveraging the creativity of a VLM. By evaluating on 12 different simulation tasks, we demonstrate the capability to design and use tools to solve challenging robotic manipulation problems. Our results show that VLMgineer outperforms baselines that either use no tools or take design specifications directly from humans. We demonstrate that our co-designed tools and actions successfully transfer to a real robot, solving three representative tasks with high performance. Our ablation studies further confirm that our evolutionary framework is a critical component, providing significant performance boosts over unstructured sampling.

**Limitations and Future Works.** While VLMgineer demonstrates significant advancements, several limitations remain: (1) Although we have shown successful sim-to-real transfer, broader validation across more dynamic real-world scenarios is needed. (2) Robot actions are represented as

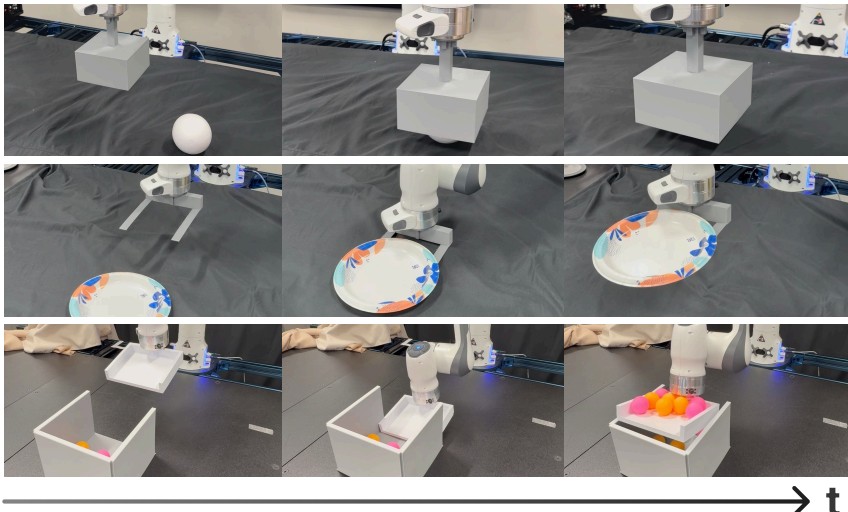

Figure 8: Snapshots of the real-world tool-action rollout on a Franka robot. The three rows correspond to three tasks: `MoveBall`, `ElevatePlate`, and `GatherSpheres`.

discrete end-effector poses, limiting the handling of complex dynamic tasks. (3) Tool representations in URDF are constrained to simple geometries, and a comprehensive evaluation of more complex, articulated tools is needed. (4) VLMGINEER is currently optimized for individual tasks, and we have not explored multitask optimization or generalization. See App. A.8 for failure modes. Future work should focus on enhancing action representations, exploring richer tool designs, and pursuing multitask generalization. (5) Current VLMGINEER pipeline does not consider manufacturing considerations and limitations. Future work needs to explore ways of incorporating these constraints into the optimization for designs that could be more readily manufactured. (6) One might ask whether it is practical to design new tools for every task. However, we cannot assume that a convenient tool already exists in the environment for every new task. Instead, the ability to design and fashion new tools suited to a task is a general capability provided by VLMGINEER. We acknowledge that integrating existing or previous tools is beneficial, and future integration with methods capable of selecting between existing and newly generated tools could enhance practicality and scalability. (7) VLMGINEER requires low-level and environment-specific information, including raw environment code and task description as input. While many other works have attempted to build digital twins of the real world (Torne et al., 2024) or training policies in simulation that are transferable to the real world (Ma et al., 2023), we instead investigate a much less studied aspect: automating the design of tools and how to wield them. Our focus is on designing a fully autonomous framework for this co-design problem, minimizing any task-specific manual design. We leave integrating well-established modules, such as digital twin constructions and sim2real policy training, to future work.

## 8 REPRODUCIBILITY STATEMENT

We aim to make our results straightforward to replicate. The benchmark tasks, environment details, and dense reward definitions are specified in A.2. The major evaluations are performed in a widely available simulator, PyBullet. The full algorithmic pipeline is illustrated in Figure 2 and described in Algorithm 1, with hyperparameters summarized in Appendix A.3.6. We include the complete set of prompt templates and composition rules used in all experiments in Appendix A.4. Finally, we indicate in the paper that the benchmark and code will be released to facilitate future research.

### ACKNOWLEDGMENTS

We thank our anonymous reviewers, who provided thorough and fair feedback that improved the quality of our paper. This work was supported by NSF CAREER Award FRR-2443721, NSF CAREER Award 2239301, NSF SLES 2331783, ONR N00014-22-1-2677, and DARPA TIAMAT HR00112490421.

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

## A APPENDIX

### A.1 BASELINE DETAILS

#### A.1.1 HUMAN-PROMPTED DESIGNS EXPERIMENT IMPLEMENTATION

Each participant underwent the following experimental procedure for each task: (i) We provided a screenshot of the environment and a description of the task, accompanied by a brief Q&A session to ensure the participants understood the task. (ii) Participants then had five minutes to write a prompt in English specifying their desired tool design and robot action. We instructed participants to be as descriptive as possible while focusing on both the design of the tool and how the robot should use it to accomplish the task. (iii) we integrated their prompt into our standardized request to the VLM (by adding instructions as shown in Appendix A.4.9), generating 5 tool described in URDF format along with a batch of 10 samples of action waypoints for each tool. (iv) The VLM outputs were then evaluated in our simulation environment using the same reward metrics described in Section 6. (v) Finally, we evaluated and recorded the best-performing tool and action pair based on the task reward metric for each participant. For a case study, we obtained prompts from three humans coming from three different backgrounds, including an LLM expert (a student with extensive research experience in LLM), a robotics expert (a student with extensive research experience in robotics), and a layperson (with no technical background). This case study will serve as an initial attempt on the concept. In the future, we plan to recruit more human subjects to conduct human study experiments on a larger sample population.

#### A.1.2 NO-TOOL EXPERIMENT IMPLEMENTATION

In the no-tool baseline experiment, we evaluate the robot's performance without any additional tool attachment. The Franka Panda robot uses its original two-finger gripper to perform the task, with the VLM generating action waypoints for the robot end effector pose and gripper open/close, totaling 7 degrees of freedom. The prompt for this baseline is adapted from our proposed prompt by removing the tool design component and associated instructions, while retaining the task description and action generation requirements. We use 5 agents with each generating 10 samples of action waypoints, evaluated using the same metrics introduced in Section 6. The complete no-tool prompt is provided in Appendix A.4.8.

#### A.1.3 RLBENCH EXPERIMENT IMPLEMENTATION

In the RLBench experiment, we evaluate the robot's performance with tools from RLbench. We assume the tool is already attached to the end effector without considering the picking step. The tool are scaled to adapt to our tasks which are similar to the ones in RLBench. The prompt for this baseline is also adapted from our proposed prompt by removing the tool design component and associated instructions. We use 5 agents with each generating 10 samples of action waypoints, evaluated using the same metrics introduced in Section 6. The complete no-tool prompt is provided in Appendix A.4.10.

### A.2 ROBOTOOLBENCH DETAILS

In this section, we provide detailed descriptions of each task and their corresponding dense reward functions in ROBOTOOLBENCH.

**BringCube**

In this task, a red cube on the desk which is out of the
reach of the robot is needed to be brought closer to the
target zone.
The reward measures how close the cube is to the target as
a fraction of its starting distance, and scales it to 0~1.

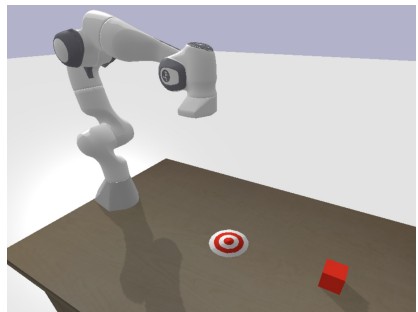

**CleanTable**

In this task, the colorful cubes representing dusts need to
be pushed away from the robot into a circular target zone
marked by the green boundary.
The reward reflects, on average, how far each cube has
been pushed toward the goal circle, and scales it to 0~1.

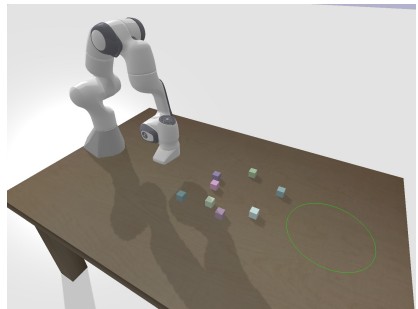

**DislodgeCube**

In this task, a red cube is confined within a white,
transparent pipe in front of the robot, which has two exits:
one opening faces the robot (along negative X) and the
other at the front-right corner (along negative Y). The
objective is to dislodge the cube through either opening.
The reward captures the cube's progress toward either of
the two pipe exits by computing two separate, normalized
(on a 0~1 scale) "distance-to-exit" scores and then taking
the better one.

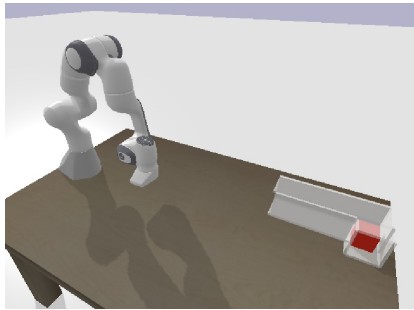

**ElevatePlate**

In this task, a white plate placed on the desk in front of the
robot needs to be securely lifted up.
The reward measures how far the plate has moved from its
starting position to the desired lifted position, and scales it
to 0~1.

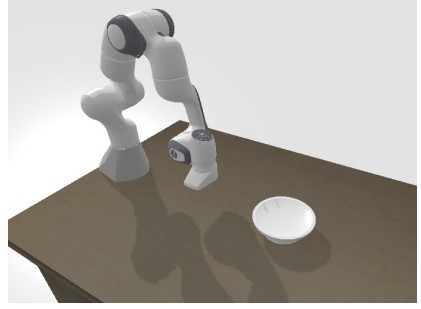

**GatherSpheres**

In this task, an open three-walled container filled with small purple spheres is placed before the robot. The objective is to gather and elevate as many spheres as possible above 0.3 m.
The reward captures, on average, how high the spheres have been lifted up to a specified cap, and scales it to 0~1.

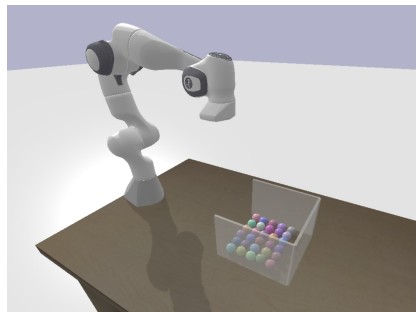

**HighObject**

In this task, a green cube sits on the top shelf. The objective is to place it inside the beige box positioned between the shelf and the robot.
The reward combines a hard "in-box" check with a smooth distance-based signal and a bonus for lowering the cube off the shelf, and scales it to 0~1.

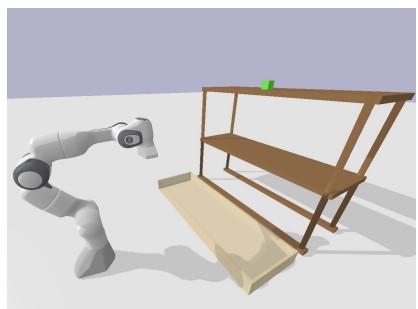

**LiftBox**

In this task, a brown box on the desk in front of the robot must be lifted above a height threshold of 0.25 m.
This reward measures how much the box has moved toward its target (lifted) position, and scales it to 0~1.

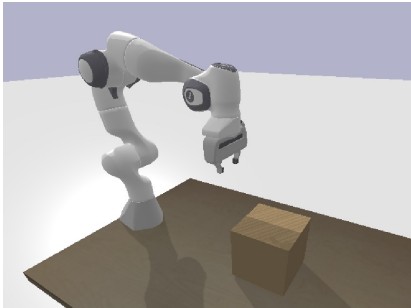

**MoveBall**

In this task, a red ball on the desk must be moved from the robot's left side to its right side.
The reward balances two objectives, getting the ball toward the right-side target and keeping its speed in check, and scales it to 0~1.

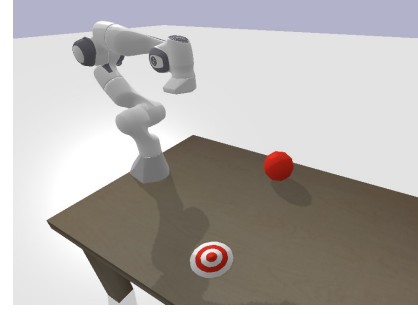

**OneBook**

In this task, two book holders with five books between them are in front of the robot. The objective is to pull out the middle (3rd) book while keeping the others in place. This reward balances two goals, pulling out the middle book and keeping the others perfectly still, and scales it to 0~1.

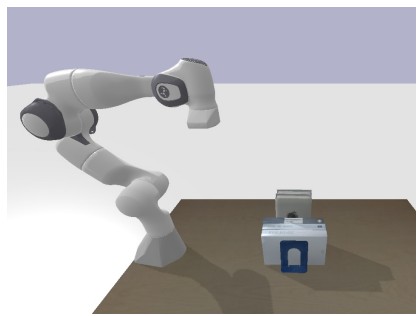

**ScoreGoal**

In this task, a hockey puck and a goal are placed on the ground far from the robot. The objective is to place the puck inside the goal.
The reward gives full credit once the puck is entirely inside the goal's 3D bounding box, and otherwise scales linearly with how much closer the puck is, horizontally, to the goal than it was at the start, and scales it to 0~1.

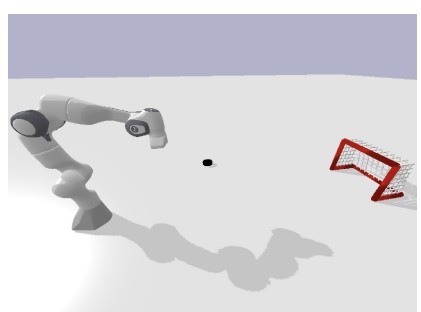

**SnatchCookie**

In this task, a transparent jar of cookies sits on the desk in front of the robot. The objective is to take at least one cookie from the jar.
The reward checks whether any cookie has been lifted out of the jar, and otherwise gives partial credit, from 0 to 1, based on how high the tallest cookie has been raised.

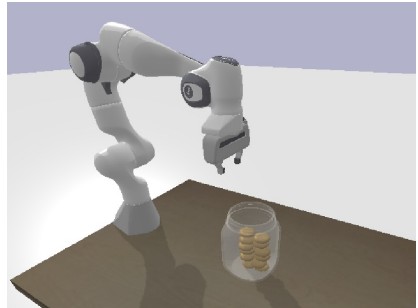

**TurkeyLegs**

In this task, a silver pot with handles on both sides, full of turkey legs, sits on the desk in front of the robot. To the pot's left (robot's perspective) is a chef's box. The objective is to transfer all turkey legs into the box without moving the pot.
The reward combines two checks, keeping the pot out of the box and getting each turkey leg into the box, by multiplying, and scales it to 0~1.

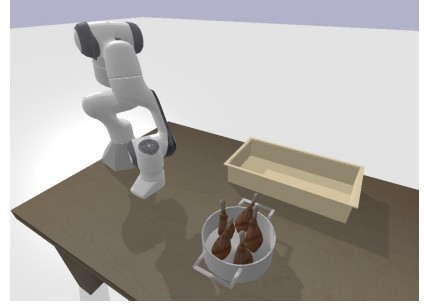

### A.3    VLMGINEER IMPLEMENTATION DETAILS

#### A.3.1    DESIGN AGENTS CONTEXT

This section describes the context that applies to a single design agent. The design agent is provided with task context as illustrated in Fig. 2, which includes (1) the environment code, (2) a screenshot of the environment, (3) a brief task description, and (4) a total text prompt composed by the prompts in Appendix A.4. We also provide task-agnostic context, including (i) environment base class, which provides basic task-agnostic functionalities, (ii) environment runner, which establishes the context for how we will use the output tool and action, and (iii) a URDF of the Franka Panda without the tool to indicate where to attach the tool. All task environments are implemented as child classes of the environment base class, ensuring they inherit the fundamental functionalities while allowing for task-specific implementations.

#### A.3.2    DESIGN AGENTS QUERIES

In VLMGINEER, we query VLM for designs by first initializing $n_{agents}$ number of agents in parallel with the same prompts. For each agent, we prompt it to generate $n_{tool}$ number of tool designs, and $n_{action}$ number of action waypoint samples that correspond to *each* tool design. Therefore, the total number of tool-action pairs that are generated via one complete query is $n_{agents} \times n_{tool} \times n_{action}$. The prompt we use to specify this behavior to each agent is presented in Appendix A.4.2.

We explicitly choose this style of querying to maximize time efficiency and design diversity: (1) time efficiency is achieved by reducing the querying algorithmic complexity by using parallel VLM agents. (2) Empirically, we found that design diversity is achieved when we balance dependence and independence between design decisions. Specifically, when a single VLM agent auto-regressively generates $n_{tool} \times n_{action}$ tool-actions pairs, having later design outputs be conditioned on previous design outputs can encourage diversity within that conditional distribution. However, in order to sample from many distinct conditional distributions, as this provides additional diversity, we found that parallel VLM queries that share no history can help with that. Ultimately, we found that optimizing time efficiency and design diversity led to better and faster initial samples as well as evolutions.

See Appendix A.3.6 for details on the values we used for these parameters for benchmarking.

#### A.3.3    DESIGN AGENT OUTPUTS

As a part of our prompt to the VLM to query for designs, we specified our desired tool and action formats. For our tool design requirements, please refer to Appendix A.4.3 for details. For our action design requirements, please refer to Appendix A.4.4 for details. Notably, these prompts are separated into with & without the Franka gripper usage. Is it also important to note that the number of waypoints the VLM outputs is completely decided by the VLM itself. During VLMGINEER's initial sampling, some design agents are asked to design tools for the gripper, some are not. This ensures the full capabilities of the default morphology are used. The two types of tools are also specified with different required attachment locations: gripper-using tools are asked to be attached to the two Franka gripper fingers, and non-gripper-using tools are asked to be attached to a "virtual joint", which is a joint we set up positioned at the flange of the Franka end effector to make the attachment process more standardized.

#### A.3.4    SIMULATION EVALUATION

From the previous section, we obtain a list of tool-action pairs in the form of URDF designs and action waypoints, respectively. To use these for simulation evaluation, we first merge the tool URDF without modification into a blank Franka Panda URDF (a blank Franka URDF will contain a gripper if gripper usage is enabled, and otherwise will not). For the action waypoints, which are inherently sparse, we implement linear interpolation for the position trajectory and SLERP (Spherical Linear Interpolation) for the orientation trajectory. The Pybullet simulation then executes these interpolated actions in the designated task environment. Finally, the environment returns result metrics for each run with the corresponding samples, allowing for both choosing evolution candidates and for producing the quantitative evaluation of the design performance. To speed up the evaluation, $k_{sim}$ samples are evaluated in parallel Pybullet simulations at a time.

A.3.5    EVOLUTION

After evaluating all previous tool-action pairs, we perform selection as follows: (1) For every task, we define two parameters to control the behavior of selection: $reward_{save}$ and $k_{top}$. (2) Using these parameters, we first select the $k_{top}$ number of tool-action pairs with the highest task rewards, and then keep only the pairs that have a reward higher than the $reward_{save}$ threshold, resulting in a set of winner tool-action pairs. We found this selection mechanism empirically allows for the best signals for evolution.

We then take this winner tool-action pair set and feed it as context into the next design agent query. These previous designs are introduced to the VLM by the "evolution mission introduction prompt" in Appendix A.4.1, where the VLM is asked to perform mutation and crossover on the previous tools via the rules specified in A.4.7. These evolved design samples will be fed into the simulation for evaluation, and the cycle will continue. We define a final $n_{iteration}$ parameter to control the number of iterations that this cycle would go on for.

See Appendix A.3.6 for details on the values we used for these parameters for benchmarking.

A.3.6    VLMGINEER BENCHMARKING DETAILS

Table 2: Benchmarking Parameters for Different Tasks

| Task Name | $n_{agent}$ | $n_{tool}$ | $n_{action}$ | $k_{top}$ | $reward_{save}$ | $n_{iteration}$ | $k_{sim}$ |
|---|---|---|---|---|---|---|---|
| BringCube | 20 | 10 | 10 | 5 | 0.6 | 3 | 100 |
| CleanTable | 20 | 10 | 10 | 5 | 0.6 | 3 | 100 |
| DislodgeCube | 20 | 10 | 10 | 5 | 0.6 | 3 | 100 |
| ElevatePlate | 20 | 10 | 10 | 5 | 0.6 | 3 | 100 |
| GatherSpheres | 20 | 10 | 10 | 5 | 0.6 | 3 | 100 |
| HighObject | 20 | 10 | 10 | 5 | 0.5 | 3 | 100 |
| LiftBox | 30 | 15 | 15 | 5 | 0.1 | 3 | 100 |
| MoveBall | 20 | 10 | 10 | 5 | 0.6 | 3 | 100 |
| OneBook | 20 | 10 | 10 | 5 | 0.4 | 3 | 100 |
| ScoreGoal | 20 | 10 | 10 | 5 | 0.4 | 3 | 100 |
| SnatchCookie | 5 | 5 | 5 | 5 | 0.3 | 3 | 100 |
| TurkeyLegs | 30 | 10 | 15 | 5 | 0.2 | 4 | 100 |

When benchmarking VLMGINEER against ROBOTOOLBENCH, we used a different set of parameters for each task, detailed in Table. 2. We used `gemini-2.5-pro-preview-03-25` as our VLM model throughout the entire experiment, and ran PyBullet evaluations on an AMD Ryzen 7 9800X3D 8-Core Processor CPU with 64 GB of RAM. On average, one run of VLMGINEER on one of these tasks should take around 30 minutes. Below, we explain each hyperparameter:

- $n_{agent}$: the number of parallel VLMgineer evolutionary design agents. Each agent independently performs design sampling and evaluation. The top-k designs across all agents are selected and passed to the next evolutionary iteration. We empirically found that employing multiple parallel agents enhances both diversity and runtime efficiency, as a single agent often becomes trapped in local minima and runs more slowly.

- $n_{tool}$: the number of tool samples generated by each evolutionary agent in each iteration.

- $n_{action}$: the number of action samples generated per tool in each iteration by each evolutionary agent. Note that tool and action samples are generated simultaneously in one pass by the VLM, resulting in a total of $n_{tool}$ x $n_{action}$ samples per iteration per evolutionary agent.

- $k_{top}$: the number of top-performing samples selected from all samples generated by all agents at the end of each iteration.

- $reward_{save}$: in addition to $k_{top}$, we apply a minimum reward threshold to determine whether a sample should be retained.

- $n_{iteration}$: the total number of evolutionary iterations conducted.

- $k_{sim}$: the number of simulation evaluation steps performed for each sampled design.

Empirically, we observed that using too few agents constrained the diversity of generated tools. Thus, we consciously selected a sufficiently large number of agents within our computational budget. For parameters such as $k_{\text{top}}$ and $reward_{\text{save}}$, our experiments indicated that selecting too few top samples reduces diversity. We further present experimental ablations to illustrate the sensitivity of VLMgineer's performance to variations in $n_{\text{action}}$ and $n_{\text{tool}}$ samples.

Our analysis reveals that VLMgineer's performance consistently improves as the number of generated samples increases. The table below reports the average reward (0.0-1.0) and standard deviation across 4 benchmark tasks for varying sample sizes. Number of samples corresponds to $n_{\text{agent}} \times n_{\text{tool}} \times n_{\text{action}}$.

## A.4 FULL PROMPTS

In this section, we provide all VLMGINEER prompts. We show individual prompt components in section A.4.1- A.4.7. We then describe we compose these prompts for different experiments in section A.4.8-A.4.10. We also include a environment code sample input in section A.4.11. For details of their usage, please refer to Appendix A.3.

### A.4.1 MISSION INTRODUCTION

Initial sampling mission introduction prompt:

```
You are a robotics hardware and controls expert.  You operate with
boldness and brilliance in the physical realm.  You work with a
robot arm that sits in the origin of your environment.  You will be
presented with some robotic tasks, and will be asked to design tools
and actions to complete the task.  Your goal is not to complete the
task to perfection in one fell swoop.  Instead, your meta-goal is
to generate a wide range of differentiated good solutions over time,
where one of them will inevitably succeed.
```

Evolution mission introduction prompt:

```
You are a robotics hardware and controls expert.  You operate with
boldness and brilliance in the physical realm.  The goal is to
create tools and actions to complete a given task.  You will be
given a list of previously generated tool designs via JSON with
URDF. Your goal is to evolve the tool designs via mutation and
crossover, and generate the new best actions for the evolved tools.
This will be done in a way that is similar to genetic algorithms,
and will be specified in detail in the "Evolutionary Process"
section below.
```

A.4.2 PROCEDURE INSTRUCTION

```
The procedure you will follow:
1.  Receive Environment Descriptions:  The user will provide some
detailed environment descriptions, robotic task instructions, and an
initial image of the workspace area from the overhead camera.
2.  Describe the Scene:  Analyze the environment.  Write down the
spatial relationship, including by not limited to the position,
orientation, dimension, and geometry of all the objects in the
scene.  Use all the information provided to you, including all text,
code, and images.
3.  Create Strategies and Designs:  You will need to create $n_{tool}$
tool that you can use to complete the task.  For each of the tools
you designed, you must generate $n_{action}$ set of action waypoints that
you can use to complete the task.  Specifically, for a total of $n_{tool}$
times, do the following steps:

     (a) First, write down a completely different, out-of-the-box
     tool design to tackle the task.  Make it unlike any other
     tool design you made in your other strategies.

     (b) Create these tools following the "Tool Specification"
     section below.

     (c) For this tool, write the following down:  (1) The spatial
     relationship (pose transformation) between the end-effector
     and each component of the tool; (2) The 3D space that each
     tool component will take up when connected to the robot; (3)
     The usage of each component of the tool when carrying out the
     task.

     (d) Use your previous analysis to tweak any obvious issues
     with the position, orientation, and dimension of your tool
     design.

     (e) Next, using your knowledge of the tool and your in depth
     analysis regarding the intricate 3D spatial relationships
     between the tool and its environment, create $n_{action}$ number of
     different step by step action plans to enable to effective
     tool use (See more in "Desired Action Criteria Definitions").
     Be very wary about how objects interact with each other

     (f) Transform your step-by-step action plan into waypoints
     adhering to the "Action Specifications".  During this
     transformation, think about the inherent nature of
     controlling robots with waypoint control and the difficulty
     that may present.
```

A.4.3 TOOL SPECIFICATIONS

Tool specification prompt without the use of Franka Grippers:

```
(Tool Specifications) Your design of the tool must follow these
rules:  (1) You must only use 3D rectangles for each component; (2)
Your tool will be outputted in a URDF block format, which should be
directly added to the end of a panda URDF file, before the robot
closing declaration; (3) Make sure your tools weigh very little in
the URDF file, where each tool part should weigh no more than a few
grams (these weights do not have to be realistic, it is just for
the robot inverse kinematics to have a easier time converging).  (4)
Your design will be a single rigid tool, which should be attached
directly to the "panda_virtual" link, which you can safely assume to
have the same orientation as the world frame.  (5) Any attachments
you design should geometrically be directly connected to their
parent links in the URDF (there should be no gaps in between!)  (6)
As a general observation, you perform better when the tools you
design are complex and intricate.
```

Tool specification prompt with the use of Franka Grippers:

```
(Tool Specifications) Your design of the tool must follow these
rules:  (1) You must only use 3D rectangles for each component;
(2) Your tool will be outputted in a URDF block format, which
should be directly added to the end of a panda URDF file, before
the robot closing declaration; (3) Make sure your tools weigh very
little in the URDF file, where each tool part should weigh no more
than a few grams (these weights do not have to be realistic, it
is just for the robot inverse kinematics to have a easier time
converging).  (4) Your design will be a pair of attachments to
the robot gripper fingers (which allows the tool to be actuated
with the robot gripper); You should attach the left attachment to
"panda_leftfinger" and the right attachment to "panda_rightfinger".
(5) Any attachments you design should geometrically be directly
connected to their parent links in the URDF (there should be no gaps
in between!)  (6) As a general observation, you perform better when
the tools you design are complex and intricate.
```

### A.4.4  ACTION SPECIFICATIONS

Action specification prompt without the use of Franka Grippers:

```
(Action Specifications) Your tool-using action will be a Nx6 numpy
array of action waypoints, where N is the number of waypoints, and
each waypoint is of dimension 6 (xyz position + roll-pitch-yaw euler
angle orientations).  Your action needs to be precisely six numbers
per waypoint.  Your waypoints will be carried out by the EnvRunner
class.  It is important to stress this:  the action waypoints are
controlling the robot end-effector "panda_virtual" link:  this
means you have to carefully take into account the dimensions of
the tool and the thickness of its parts when designing effective
waypoints.  Again, you can safely assume the end-effector has the
same orientation as the world frame upon initialization (see frame
clarification again for details)!
```

Action specification prompt with the use of Franka Grippers:

```
(Action Specifications) Your tool-using action will be a Nx7 numpy
array of action waypoints, where N is the number of waypoints, and
each waypoint is of dimension 7 (xyz position + roll-pitch-yaw euler
angle orientations + binary gripper open/close state in integers
[0 for open, 1 for closed]).  Your action needs to be precisely
seven numbers per waypoint.  Your waypoints will be carried out by
the EnvRunner class.  It is important to stress this:  the action
waypoints are controlling the robot end-effector "panda_virtual"
link:  this means you have to carefully take into account the
dimensions of the tool and the thickness of its parts when designing
effective waypoints.  Again, you can safely assume the end-effector
has the same orientation as the world frame upon initialization (see
frame clarification again for details)!
```

### A.4.5   ACTION DIVERSITY SPECIFICATION

```
(Desired Action Criteria Definitions) For the description below, we
will call a single sequential set of waypoints in a single rollout
as one "action set".  For each tool you created, the goal is to
generate $n_{action}$ action sets that optimize the task success and motion
differentiation.  Task success is optimized when an action set is
able to complete the task successfully.  Motion differentiation is
optimized when there exists a large variance in the motion taken
across all action sets you design for the same tool.  A large
variance in motion is defined the tool, at each time step, is
located at a different location in the 3D space.  Think about how
a tool can be used to interact with the object from many different
sides, angles, and ways.  When both conditions are met, you have
successfully designed a good set of actions sets.
```

### A.4.6   FRAME CLARIFICATIONS

```
(Frame Clarification) In the world frame, front/back is along the
x axis, left/right is along the y axis, and up/down is along the
z axis with the following directions:  Positive x:  Towards the
front of the table.  Negative x:  Towards the back of the table.
Positive y:  Towards the left.  Negative y:  Towards the right.
Positive z:  Up, towards the ceiling.  Negative z:  Down, towards
the floor.  In terms of orientation, starting from the origin frame,
Positive rotation about the x-axis:  tilting the end-effector head
to the left.  Negative rotation about the x-axis:  tilting the
end-effector head to the right.  Positive rotation about the y-axis:
tilting the end-effector head down.  Negative rotation about the
y-axis:  tilting the end-effector head up.  Positive rotation about
the z-axis:  rotating the end-effector head counter-clockwise.
Negative rotation about the z-axis:  rotating the end-effector head
clockwise.
```

### A.4.7 EVOLUTIONARY INSTRUCTIONS

> (Evolutionary Process) Your design decision is a part of a tool design genetic algorithm. For each of the $n_{tool}$ tool designs, you can choose to either mutate or crossover. Specifically, tool mutation is defined as one change to a single randomly selected previous tool design. Mutation changes include:
>
> > (1) Changing the dimension, location, or orientation of a single component of the tool.
> >
> > (2) Adding, removing, or replacing a single component of the tool.
>
> Crossover is defined as the process of combining two randomly selected previous tool designs to create a new tool design. Combination is defined as:
>
> > (1) Selecting components from two previous tool designs and combining them to form a new tool design.
>
> All mutation and crossover decisions must potentially increase the likelihood of task success, yet all decisions must be different and diverse.

### A.4.8 NO TOOL INSTRUCTIONS

> You are a robotics hardware and controls expert. You operate with boldness and brilliance in the physical realm. You work with a robot arm that sits in the origin of your environment. You will be presented with some robotic tasks, and will be asked to design actions to complete the task.
>
> ...

The complete prompt is composed together with instructions from A.4.2, A.4.4, A.4.5, and A.4.6.

### A.4.9 HUMAN SPECIFICATION INSTRUCTIONS

> You are a helpful robotics hardware and controls expert. You have a robot arm that sits in the origin of your environment. You are working with a colleague as a team to design tools and actions for a robot to complete a task. Your colleague will provide you with a design and action instructions in the form of natural language instructions. Your goal is to use your colleague's design and action instructions to output URDF and action waypoints for the robot to use. You should not use your own knowledge to design the tool and action, but rather follow your human colleague's instruction. Here is the human colleague's prompt: {human_prompt}
>
> ...

The complete prompt is composed together with instructions from A.4.2, A.4.3, A.4.4, A.4.5, and A.4.6.

### A.4.10   RLBench Instructions

```
You are a helpful robotics hardware and controls expert.  You have
a robot arm that sits in the origin of your environment.  You are
working with a colleague as a team to design tools and actions
for a robot to complete a task.  Your colleague will provide you
with a design in the format of a URDF, which is attached for you as
tool.txt.  Your goal is to use your colleague's URDF to come up with
an action plan for the robot to use.

...
```

The complete prompt is composed together with instructions from A.4.2, A.4.4, A.4.5, and A.4.6.

### A.4.11   Sample Environment Code

```python
from envs.base_env import BaseEnv
from models.primative_objects.cube import Cube
import numpy as np
import pybullet as p

class BringCubeCloserEnv(BaseEnv):
    def __init__(
        self,
        **kwargs,
    ):
        self.start_cube_pos = np.array([1.0, 0, 0.035])
        self.target_cube_pos = np.array([0.6, 0, 0.035])

        super().__init__(
            **kwargs,
        )

    def _setup_environment(self):
        # Load plane
        planePos = [0, 0, -0.625]
        planeOri = p.getQuaternionFromEuler([0, 0, np.pi/2])
        self.plane = p.loadURDF("plane/plane.urdf", \
        planePos, planeOri, useFixedBase=True)
        p.changeDynamics(self.plane, -1, restitution=0.95)

        # Load table
        tablePos = [0.6, 0, -0.625]
        tableOri = p.getQuaternionFromEuler([0, 0, 0])
        self.table = p.loadURDF("table/table.urdf", \
        tablePos, tableOri, useFixedBase=True)

        # Load cube
        cubeSize = 0.07
        self.cube = Cube(cubeSize, self.start_cube_pos).get_shape()

        # Load visual goal
        self.goal = p.loadURDF("visual_goal/goal.urdf", \
        self.target_cube_pos-np.array([0, 0, 0.035]), useFixedBase=True)

        if self.blender_recorder:
            self.blender_recorder.register_object(self.table, "table")
            self.blender_recorder.register_object(self.cube, "cube")
            self.blender_recorder.register_object(self.goal, "goal")

    def reward(self):
        cube_pos, _ = p.getBasePositionAndOrientation(self.cube)
        current_distance = np.linalg.norm(self.target_cube_pos - \
        np.array(cube_pos), ord=1)
        initial_distance = np.linalg.norm(self.target_cube_pos - \
```

```
50            self.start_cube_pos, ord=1)
51
52            # Normalize reward to be 0 at initial position and 1 at target
53            normalized_reward = max(0, 1-current_distance/initial_distance)
54            return normalized_reward
55
56        def reset(self):
57            super().reset()
58            # Reset cube position
59            p.resetBasePositionAndOrientation(
60                self.cube,
61                self.start_cube_pos,
62                p.getQuaternionFromEuler([0, 0, 0])
63            )}
```

## A.5 TOOL DESIGN GALLERY

In this tool design gallery, we take the opportunity to display tools from a few tasks that seemed to have allowed VLMGINEER the most creative freedom. These are tool designs that are not presented elsewhere in the paper. We believe this illustrates VLMGINEER's impressive physical creativity and problem-solving capabilities.

| Task Name | | | |
|---|---|---|---|

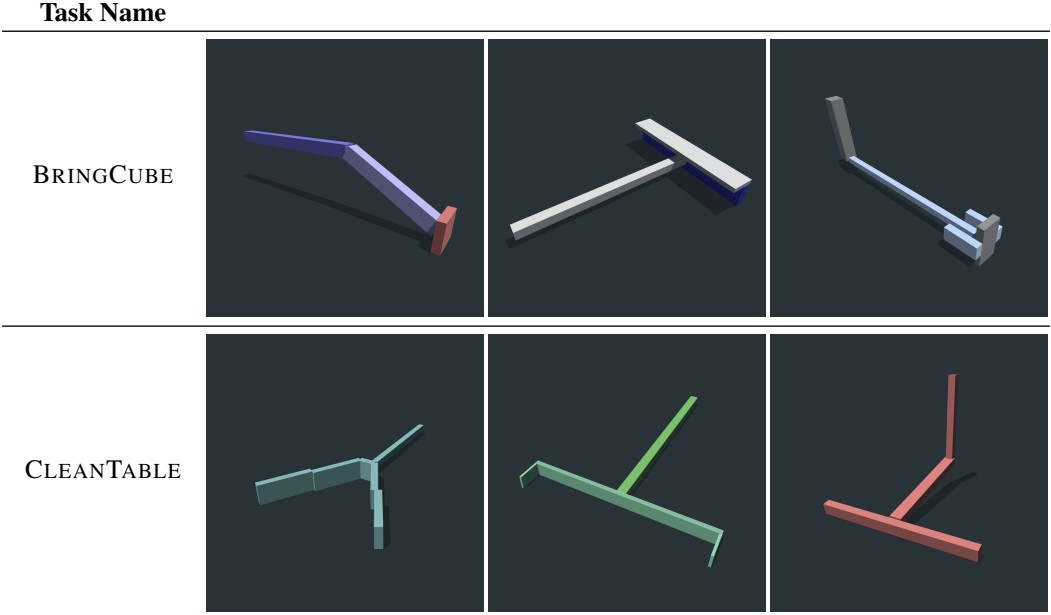

BRINGCUBE

CLEANTABLE

| Task Name | | | |
|---|---|---|---|

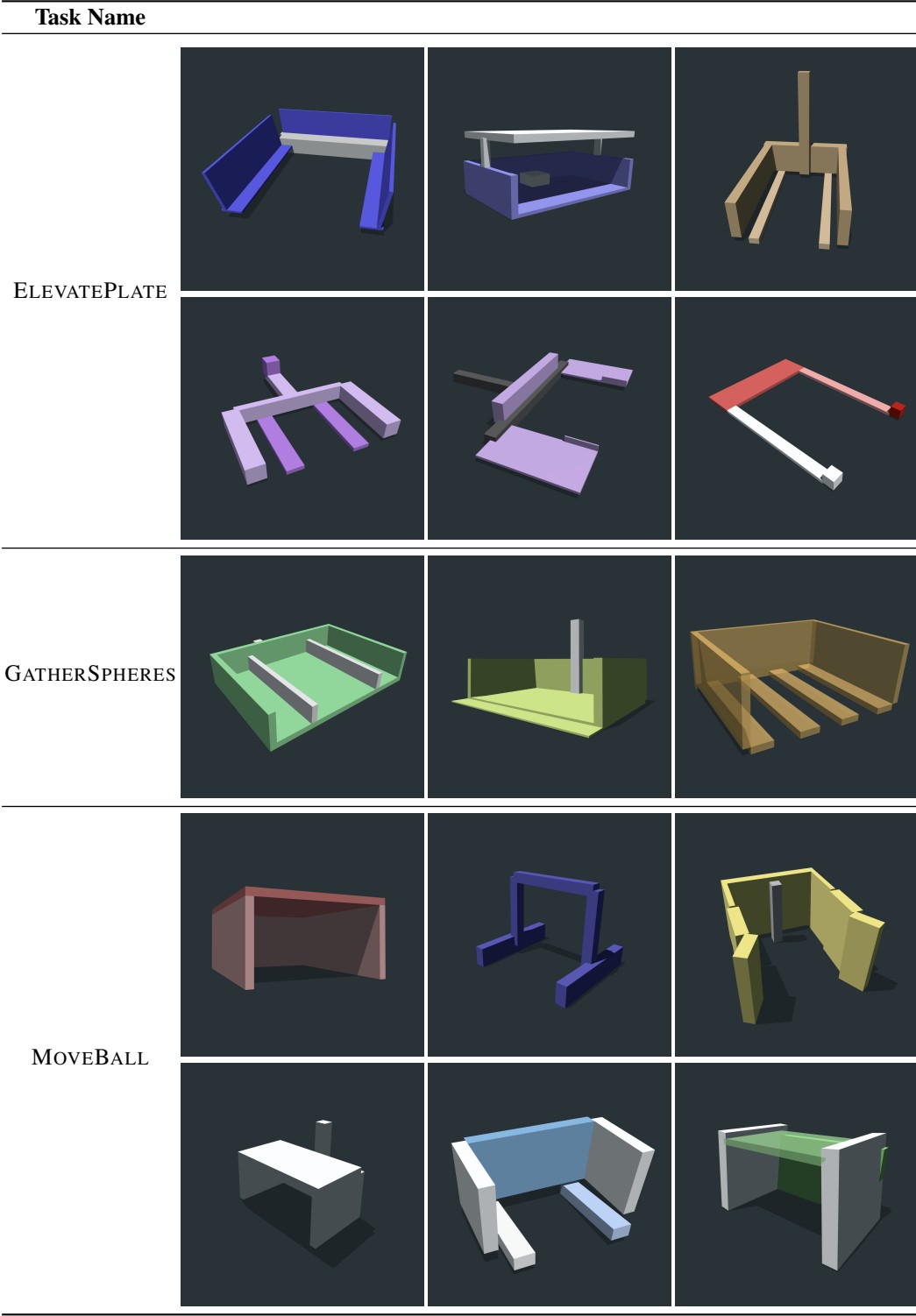

ELEVATEPLATE

GATHERSPHERES

MOVEBALL

## A.6 AN EXAMPLE OF ACTION SEQUENCES

Each action set is a 6-(or 7-)dimensional vector representing [x, y, z] position and [roll, pitch, yaw] orientation (and an optional gripper command). The number of action waypoints is not fixed; it varies

across trajectories and is fully determined by the VLM, not by any preset specification. An example is shown below.

```
"action_sets": [
    [
        [ 0.6, 0.0, 0.4, 0.0, 0.0, 0.0 ],
        [ 0.6, 0.0, 0.32, 0.0, -0.2, 0.0 ],
        [ 0.8, 0.0, 0.32, 0.0, -0.2, 0.0 ],
        [ 0.8, 0.0, 0.32, 0.0, 0.2, 0.0 ],
        [ 0.8, 0.0, 0.7, 0.0, 0.2, 0.0 ]
    ],
```

## A.7 STATISTICAL SIGNIFICANCE ANALYSIS

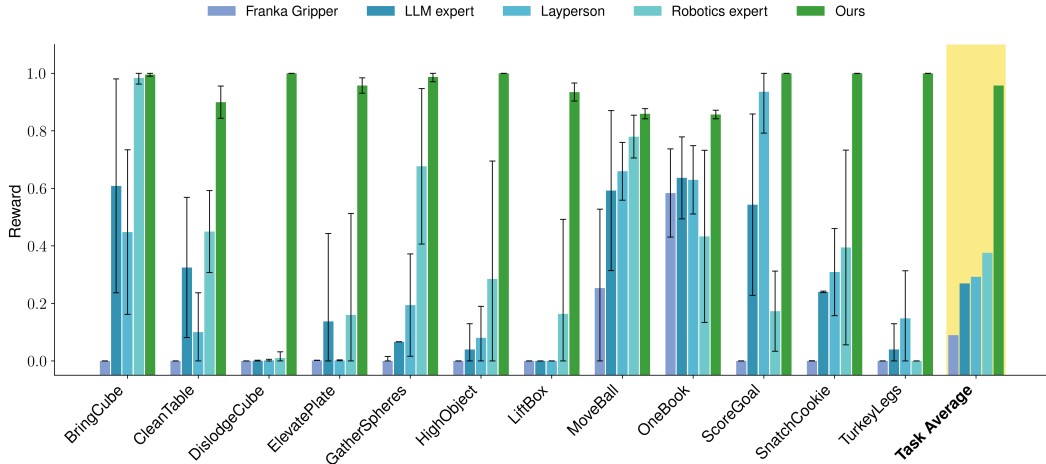

Figure 9: Comparison of the mean and standard deviation of reward generated by VLMGINEER, human-prompted designs, and Franka Gripper across 12 tasks. Error bars represent standard deviation (clipped to the range $[0, 1]$).

Fig. 9 presents our primary quantitative results, including standard deviations across 5 runs. Across all 12 tasks, VLMGINEER consistently surpasses all baselines, exhibiting notably low variation between trials. This indicates that VLMGINEER reliably produces high-performing and stable tool-action designs.

In contrast, results obtained from human-prompted designs not only yield significantly lower performance but also show greater variations across runs. We attribute this discrepancy to several factors. First, human-specified tools often require more intricate control strategies; even if capable of completing the task, these designs tend to be less resilient to suboptimal or imperfect executions. By comparison, VLMGINEER-generated tools typically exhibit greater robustness to action imperfections. Second, human prompts sometimes suffer from specification ambiguity or misalignment with the VLM. There can be discrepancies between human intent and the VLM's internal representation and physical modeling capabilities. By automating the design process, VLMGINEER avoids these alignment issues, resulting in more effective and precisely realizable solutions.

## A.8 VLMGINEER'S COMMON FAILURE MODES

In the VLMGINEER pipeline, common failures include physically infeasible initial tool designs. For example, generated tools sometimes penetrate the environment due to inaccurate simulation physics or intersect with the robot itself at the beginning of the task. More careful prompting and post-processing can help mitigate these. Furthermore, additional practical failure modes emerge when attempting real-world deployment, such as tools being too thin for reliable 3D printing and excessively large for single-piece fabrication.

## A.9 SIM-TO-REAL TRANSFER PROCESS

We selected three simulated task environments in the RoboToolBench, MoveBall, ElevatePlate, and GatherSpheres, as the basis of this real-world experiment. We selected these three environments for how easy they would be to construct their real-world equivalent. For each task in the simulation, we would first attach our custom-made Franka end-effector mounting head to the robot at the location where the tools were specified to be mounted via our prompt. This allows our tool to be directly integrated with the mount, which makes real-world tool attachment simple. After getting a winning candidate, we inspect the design and trim it using a bounding box. The bounding box will remove the portions that overlap with the attachment. After that, we run a script to automatically convert the mount-attached tools to a single URDF file and convert it directly to a printable STL file using Blender. We import the STL file into our 3D printer's software (Bambu Lab), perform typical 3D printing processing (placing, adding support, slicing), and directly send the job to the 3D printer (Bambu Lab P1S). After printing and removing the supports (if present), since our tools have mounts attached, we can directly attach the tool to our Franka Panda robot end effector via screws. With the respective real-world task environment already set up, we simply send the VLMgineer optimized end-effector action waypoints to our Franka position controller for execution. After the action trajectory is carried out, we measure the relevant object states to determine the reward obtained by this tool and action tuple.

## A.10 COMPARISON OF DIFFERENT VLMS

In Table 4 and Table 5, we provide the performance comparison across different VLM models. Table 4 compares Gemini-2.5-pro and GPT-o3 on average and best-run reward, and it shows that Gemini-2.5-pro clearly leads over GPT-o3; Table 5 compares the performance across Gemini family, which shows that 2.5-pro is the strongest one, while 2.5-flash is mid-tier and 2.0-flash trails, and the step from 2.5-flash to 2.5-pro yields obvious gains.

Table 4: Performance Comparison Across Different VLM Models.

| Model | Avg. Reward | Top Reward |
|---|---|---|
| Gemini-2.5-pro | **0.6054** | **0.8222** |
| GPT-o3 | 0.3775 | 0.5436 |

Table 5: Performance Comparison Across the Gemini family.

| Model | Avg. Reward | Top Reward |
|---|---|---|
| 2.5-pro | **0.6054** | **0.8222** |
| 2.5-flash | 0.3393 | 0.4481 |
| 2.0-flash | 0.0686 | 0.0796 |

## A.11 COMPARISON OF IMAGE FEEDBACK

To evaluate our design choice of using purely the tool-action reward signal for evolution, we ablated our method against a variation of our method that added each tool-action execution's last frame to the next VLM evolution query as feedback signal. We tested this method variation on three tasks, *ElevatePlate, GatherSpheres, and MoveBall,* running this and the baseline on 500 samples per iteration with 3 evolution iterations. Note that this is in total a smaller sample size than our main experiment, which used about 2000 samples per iteration. We see in Table 6 that the inclusion of visual feedback leads to an small decline of 5.4% in average reward. This quantitative result supports our qualitative observation that VLMs often struggle to accurately ground fine-grained physical progress from raw visual observations, leading to noisy selection signals that can hamper the evolutionary search.

## A.12 COMPARISON OF SINGLE-SAMPLE ITERATIVE REFINEMENT

To evaluate the effectiveness of our population-based evolutionary search, we compare against a simple iterative refinement baseline that represents a natural alternative approach for VLM-driven

Table 6: Ablation study on feedback modality: Comparison of VLMgineer with and without image feedback across three tasks.

| Method | Elevate Plate | Gather Spheres | Move Ball | Average |
|---|---|---|---|---|
| VLMgineer w. Img Feedback | **0.30** | 0.45 | 0.80 | 0.52 |
| VLMgineer w/o. Img Feedback (Ours) | 0.28 | **0.56** | **0.82** | **0.55** |

tool action co-design. In this baseline, the VLM maintains a single tool design at each iteration rather than a diverse population of candidates. At each step, the VLM receives the current tool design, its associated action sequence, and the scalar reward feedback same as in the proposed evolutionary approach. The VLM then directly refines the design based on this feedback, generating an improved tool and N=10 action plan. Unlike our approach, which leverages population diversity through mutation and crossover operators to explore the design space broadly before converging on solutions, the iterative refinement baseline performs greedy local search from a random initial guess, producing lower performance. We tested the baseline in three tasks: *ElevatePlate*, *GatherSpheres*, and *MoveBall*, with each running refinement for 30 times. The running procedure for this baseline is exactly the same as our main proposed method. We use the same setup for our proposed method as in the image feedback ablation A.11. This experiment shows that our evolutionary approach outperforms iterative prompt refinement. Iterative refinement is easy to stuck at local minimum with bad initial proposal while evolutionary approach could expand exploration, leading to higher performance.

Table 7: Ablation study on evolutionary vs. single sample iterative prompt refinement

| Method | Elevate Plate | Gather Spheres | Move Ball | Average |
|---|---|---|---|---|
| Iterative Refinement | 0.0 | 0.07 | 0.76 | 0.28 |
| Evolutionary (Ours) | **0.28** | **0.56** | **0.82** | **0.55** |

## A.13 LICENSES

The cardboard box asset used in LIFTBOX environment is from PartNet-Mobility Dataset Xiang et al. (2020). Their terms of use are stated here: sapien.ucsd.edu/about.

The book assets and the book holder used in the ONEBOOK environment came from the YCB Dataset Calli et al. (2015). This dataset is under the CC BY 4.0 license.

The goal frame and net assets used in the SCOREGOAL environment came from the Meta-World Benchmark Yu et al. (2019). This benchmark is under the MIT License.

The transparent jar asset used in the SNATCHCOOKIE environment came from cgtrader, a 3D CAD model website. This asset is under the "Royalty Free No Ai License", detailed here.

The cookie assets used in the SNATCHCOOKIE environment came from sketchfab, a 3D CAD model website. This asset is under the CC BY 4.0 license.

The turkey leg assets used in the TURKEYLEGS environment came from sketchfab, a 3D CAD model website. This asset is under the CC BY 4.0 license.

## A.14 THE USE OF LARGE LANGUAGE MODELS (LLMS)

We mainly use LLMs to polish our writing. The use of LLMs in this work could not be regarded as a contributor.

