# OpenReview forum: "VLMgineer: Vision-Language Models as Robotic Toolsmiths"
_ICLR.cc/2026/Conference — ICLR 2026 Poster_

### Official Review · Reviewer_5kyU · 2025-10-29

**Soundness:** 3
**Presentation:** 3
**Contribution:** 3
**Rating:** 8
**Confidence:** 3

**Summary:**

This paper presents VLMgineer, a VLM-aided evolutionary algorithm that jointly optimizes tool design and action strategy to solve challenging manipulation tasks beyond the capabilities of conventional robot morphologies. It is validated that the inherent physical creativity of vision-language models, when paired with iterative evaluation and refinement, can yield interesting tool-action pairs that outperform human-designed counterparts. The paper also introduces a comprehensive simulation benchmark, RoboToolBench, for evaluating tool design performance, which could facilitate further investigation.

**Strengths:**

1.The paper addresses an important research problem, i.e. free-form evolutionary optimization of tool design to enhance the capabilities of robotic systems in challenging scenarios. An innovative VLM-based approach is presented that combines the physical creativity of VLMs with evolutionary computation to yield high-performing tool-action pairs.

2.The paper is well-written and easy to follow. The abundant illustrations and qualitative results greatly help with readers’ understanding.

3.Full prompts and implementation details are provided to ensure reproducibility. The authors also provide an actionable procedure for transferring the co-optimized solutions onto physical robot arms, increasing real-world practicality.

**Weaknesses:**

1.In the proposed method, the VLM is prompted to directly generate discrete action waypoints, instead of using reinforcement learning. From my understanding, this would more or less rely on the privileged geometric information provided in environment source codes (such as object positions and orientations). The authors could discuss in more detail the specific information provided regarding environment setup, and what is the implication and generalizability of such control strategies for less structured, real-world settings where only images and textual instructions are available.

2.The tool use setting considered in this work is somewhat simplified, where the tool is attached to the robot arm and fixed all along. Since in this case the robot morphology and tool could be considered as a whole, “augmented” morphology, the authors are suggested to discuss what are the unique challenges to be solved for this particular problem setup, compared with previous morphological design studies that also leverage large models.

**Questions:**

1.Could the authors describe how much does the proposed approach rely on the spatial reasoning capabilities of VLMs, and how much on the privileged geometric information, in order to directly generate action waypoints?

2.In many cases expert knowledge could still serve as useful informative priors. I wonder whether the authors have tried utilizing the human-specified or RLBench tools as an initial population and further evolve them with VLMgineer. Would this yield additional performance gains, or instead impose human biases that constrain more creative search?

---

> ### Author Response · Authors · 2025-11-26
> **Rebuttal by Authors**
>
> We thank the reviewer for the detailed and insightful feedback.
>
> **Weakness 1: Need for Discussion on “privileged geometry information” and real-world generalization**
>
> We believe that our approach is best interpreted in the context of sim-to-real policy transfer, which is a well-established paradigm. Therefore, similar to other work, Eureka (Ma et al. 2023\) or RialTo (Agarwal, A., et al. 2024\) in the sim2real context, we also rely on the existence of a simulation environment and, naturally, by extension, the environment code. However, while these and many other works have attempted to build digital twins of the real world or training policies in simulation that are transferable to the real world, we instead investigate a much less studied aspect: automating the design of tools and how to wield them.  Our focus is on designing a fully autonomous framework for this co-design problem, minimizing any task-specific manual design.  We leave integrating well-established modules, such as digital twin constructions and sim2real policy training, to future work. We will add a note to the paper to avoid this confusion.
>
> As for the specific information provided to the VLM, we clarify that the VLM is provided with the environment source code, a static screenshot, and a task description, as detailed in **Appendix A.3.1** and the full prompts in **Appendix A.4**. For further clarity, we will be sure to add a sample environment source code to the appendix, which would look like a gym-style python environment definition.
>
> Question 1: **How much does the method rely on spatial reasoning vs. privileged geometry?**
> We argue that even when provided privileged geometric information, the VLM would still have to spatially reason based upon given information to perform co-design. Although geometric pose data provides high-level guidance, the valid action space is highly sparse; as a result, most initial VLM-generated tools and actions are incorrect or suboptimal. This difficulty is due to privileged information provided, not including complex physical dynamics such as fine-grained object geometry, contact forces, and collision avoidance routes. The VLM must therefore implicitly reason about these unstated constraints to synthesize functional action waypoints.
>
> Weakness 2: **Simplified “tool attached to arm” setting vs. general morphology design**
> Compared to previous morphological design studies leveraging LLMs, such as *LASeR* and *RoboMorph*, our problem setting presents three unique challenges. First, *VLMgineer* requires the VLM to creatively co-design both the morphology (tools) and the associated actions simultaneously, whereas prior works focus primarily on morphology generation alone. Second, while previous studies target locomotion—a domain often defined by narrower objectives like speed and efficiency—our focus on manipulation necessitates complex reasoning about object affordances, geometry, and spatial interactions within the environment. Finally, unlike the structured and constrained design spaces of prior works, which are often amenable to traditional optimization, our approach operates within an open-ended URDF design space with technically limitless degrees of freedom and shape possibilities.
>
> Question 2: **Using human/RLBench tools as priors**
>
> Thank you for this interesting suggestion. We have not tried this because our focus has been on evaluating VLMgineer’s capability to generate solutions to novel tasks *without* task-specific priors. However, this is an interesting experiment to try. We hypothesize that limiting the initial population to only be these human-specified tools will limit the diversity of explored solutions, which could lead to worse performance before evolution requires diversity. Due to time constraints in the rebuttal period, we will aim to rigorously perform this experiment and report the results in the final version of the paper.

---

### Official Review · Reviewer_gsFY · 2025-10-30

**Soundness:** 3
**Presentation:** 2
**Contribution:** 3
**Rating:** 6
**Confidence:** 4

**Summary:**

This paper introduces VLMGINEER, an autonomous framework that uses vision–language models (VLMs) together with evolutionary search to co-design both robotic tools and action policies from scratch. Unlike prior work that assumes a fixed toolset or relies on human-specified design parameters, VLMGINEER automatically generates tool geometries (in URDF) and corresponding action waypoints, iteratively refining them through simulation-based evaluation. To benchmark this setting, the authors propose ROBOTTOOLBENCH, a suite of 12 manipulation tasks that are infeasible using a standard gripper. Experiments show that VLMGINEER consistently outperforms human-designed tools, VLM-generated tools from human prompts, and RLBench baselines, with strong improvements in both success rates and reward metrics. The framework also demonstrates successful sim-to-real transfer on a physical Franka robot.

**Strengths:**

- **Novel use of inductive in-context crossover and mutation:**
  The paper introduces an interesting concept of using VLMs to perform crossover and mutation entirely in-context without predefined operators. This is a creative and promising perspective that differentiates the approach from traditional evolutionary robotics.

- **Clear demonstration of tool-design capability:**
  The proposed framework is able to zero-shot generate both tool geometry and corresponding action strategies, showcasing VLM-based physical reasoning beyond conventional policy learning.

- **Benchmark contribution:**
  ROBOTOOLBENCH could serve as a useful benchmark for future research on tool co-design and automated physical reasoning.

**Weaknesses:**

- **Citation formatting reduces readability.**
  The manuscript embeds citations directly in narrative form, which disrupts flow and clarity. Using standard citation commands (e.g., `\citep{}`) would improve readability and maintain academic consistency.

- **Missing key related work: *RobotSmith*.**
  The paper does not reference *RobotSmith: Generative Robotic Tool Design for Acquisition of Complex Manipulation Skills*, one of the closest prior works on tool co-design. Including this would strengthen the literature coverage and contextualize the contribution.

- **Overstated novelty claims.**
  The claim that “VLMGINEER leverages the surprising physical creativity of VLMs...” suggests first-mover novelty. However, works such as *Evolution 6.0: Evolving Robotic Capabilities Through Generative Design* have similarly explored creative, generative physical design via LLMs. The authors should clarify what is genuinely new here and frame contributions accordingly.

- **Limited evaluation on long-horizon tool-use tasks.**
  ROBOTOOLBENCH primarily evaluates single-stage manipulation tasks. Including long-horizon, multi-step tool-use scenarios with meaningful constraints would better demonstrate reasoning depth and practical utility.

- **Ambiguity in Table 1 metrics.**
  It is unclear whether reported values represent normalized reward, success rate, or another metric. Clearly defining the evaluation metric and explaining its interpretation would improve transparency.

- **Lack of manufacturability considerations.**
  Although real-world results are included, manufacturability constraints are not integrated during tool generation. Some designs (e.g., for *ScoreGoal* and *BringCube*) appear unnecessarily complex compared to human-designed solutions. Incorporating manufacturability into the optimization process would improve real-world relevance.

- **Baseline design comparison could be expanded.**
  Current baselines rely primarily on human-prompted tool designs, which may introduce expertise-dependent bias and variance. Including recent LLM-based baselines—such as *RobotSmith* or simple LLM-generated tool-and-action plans without evolutionary refinement—would provide a stronger and clearer comparison, better contextualizing the advantages of the proposed method.

**Questions:**

- **Quantitative benefit of joint tool–action sampling.**
  Have you evaluated whether jointly sampling tool geometry and actions in a single VLM pass outperforms a sequential approach (i.e., generating tools first, then actions with the VLM)? Providing such a comparison would help validate this core design decision.

- **Justification for action representation.**
  Given that VLMs often struggle with spatial grounding and precise low-level control, what motivated the choice of directly predicting waypoint sequences instead of using motion primitives or parameterized action abstractions? Please clarify this design choice or include ablation experiments.

- **Separating tool-quality vs. action-quality effects.**
  Since execution relies on VLM-generated waypoints, how do you distinguish failure cases caused by suboptimal tool design from those stemming from imperfect action generation? Disentangling these factors is important for interpreting the contribution of the evolutionary pipeline.

- **Clarity in Figure 6.**
  The meaning of the light blue and light orange shaded regions is not obvious from the figure or its caption. Please provide a clearer explanation of what these regions represent.

- **Evolutionary search vs. simpler iterative refinement.**
  Have you compared your evolutionary algorithm against a simpler iterative prompting baseline (e.g., repeatedly asking the VLM to refine the tool design without population-based evolution)? Such a comparison would help isolate the specific advantages of evolutionary search.

---

> ### Author Response · Authors · 2025-11-26
> **Rebuttal by Authors**
>
> We thank the reviewer for the detailed and insightful feedback.
>
> **Weakness 2: Missing key related work: RobotSmith**
> We thank the reviewer for highlighting *RobotSmith*. *RobotSmith* is a concurrent work that appeared just one week before our submission and lacked public code availability at that time. We regard this as an impressive and significant concurrent contribution that strongly validates the emerging direction of VLM-driven mechanical design.
>
> However, our technical approaches differ significantly. In RobotSmith, VLM is used as a modular component in an otherwise conventional pipeline: it is only called a small number of times to generate an initial tool design and action, using visual reasoning as feedback. It then hands off to a conventional evolutionary optimization loop (CMA-ES) that does not continue to incorporate VLM feedback to close the loop. As a result, RobotSmith’s optimization procedure only considers scaling, rotation, and translation (as mentioned in their limitations), which restricts more complex structural changes and limits the exploration of the full design space, especially if the initial coarse design proposed by the VLM is a local minima.
>
> In contrast, in our pipeline, the VLM acts as a closed‑loop co‑designer and is engaged throughout our evolution pipeline, from diverse parallel candidate sampling to in-context crossover and mutation. Our method is able to explore a larger sample of candidate designs represented by URDF, a general tool representation format, to enable richer forms of tool modification, such as topology editing or part reconfiguration (see Figure 7). As a result, our work better leverages VLMs’ physical creativity to create diverse, unique, and unconventional designs (see Appendix A.5).
>
> \[1\] Khemlani, Sangeet, et al. "Vision language models are unreliable at trivial spatial cognition." arXiv preprint arXiv:2504.16061 (2025).
> \[2\] Wang, Junying, et al. "Affordance Benchmark for MLLMs." arXiv preprint arXiv:2506.00893 (2025).
> \[3\] Anis, Ahmad Mustafa, Hasnain Ali, and Saquib Sarfraz. "On the limitations of vision-language models in understanding image transforms." arXiv preprint arXiv:2503.09837 (2025).
> \[4\] Chow, Wei, Jiageng Mao, Boyi Li, Daniel Seita, Vitor Campagnolo Guizilini, and Yue Wang. PhysBench: Benchmarking and Enhancing Vision-Language Models for Physical World Understanding. International Conference on Learning Representations (ICLR), 2025 (Oral).
>
> **Weakness 3: Overstated novelty claims.**
> We appreciate the reviewer for bringing *Evolution 6.0: Evolving Robotic Capabilities Through Generative Design* to our attention; we will discuss and cite this work in the final revision. While prior efforts like *Evolution 6.0* have demonstrated that VLMs can generate functional tool designs, often resembling existing objects found in online databases, our core novelty lies in the finding that **evolutionary search** can elicit physical creativity from VLMs significantly beyond single-shot prompting. This process allows our system to discover nonstandard, highly performant geometries that are not merely retrieved but iteratively optimized for specific tasks. As demonstrated quantitatively in **Figure 6** and **Table 1**, our evolutionary approach yields tools that are significantly more effective than those from the initial VLM-generated population, confirming that the iterative refinement is crucial for grounding the design in physical performance.
>
> **Weakness 4: Limited evaluation on long-horizon tool-use tasks.**
> Our work intentionally focuses on co-designing tools and low-level control, not on high-level, long-horizon task planning. There is already substantial prior work on sequential task planning and symbol–skill composition—such as SymSkill \[1\] and Task and Motion Planning (TAMP) frameworks \[2, 3\]—which focus on chaining capabilities to solve multi-stage problems. Similarly, recent works in LLM-based planning like SayCan \[4\] demonstrate how to decompose long-horizon instructions into executable primitives. These planning frameworks are complementary and orthogonal to our contribution. Integrating VLMgineer with such planners is an interesting direction for future work, but it lies outside the scope of this paper.
>
> \[1\] Shao, Y., et al. "SymSkill: Symbol and Skill Co-Invention for Data-Efficient and Real-Time Long-Horizon Manipulation." arXiv preprint (2025).
> \[2\] Garrett, Caelan Reed, et al. "Integrated Task and Motion Planning." Annual Review of Control, Robotics, and Autonomous Systems 4 (2021): 265-293.
> \[3\] Simeonov, Anthony, et al. "A Long Horizon Planning Framework for Manipulating Rigid Pointcloud Objects." Conference on Robot Learning (CoRL) (2021).
> \[4\] Ahn, Michael, et al. "Do As I Can, Not As I Say: Grounding Language in Robotic Affordances." Conference on Robot Learning (CoRL) (2022).
>
> **(Continued below...)**

---

> ### Author Response · Authors · 2025-11-26
> **Rebuttal by Authors**
>
> **Weakness 6: Lack of manufacturability considerations.**
> We acknowledge the reviewer's feedback regarding manufacturability constraints. We view *VLMgineer* as the first fundamental step in a larger research agenda aimed at practical automatic tool generation. The primary scope of this work is to establish the core algorithmic framework and investigate the capacity of VLMs for physical creativity through evolutionary search, rather than optimizing for immediate industrial fabrication. However, we agree that integrating explicit manufacturability constraints is a crucial next step to improve real-world relevance, and we will highlight this as a key direction for future work in the final revision.
>
> **Weakness 7: Baseline design comparison could be expanded.**
> For our response on RobotSmith, please refer to our answer to weakness 2\.
>
> For “simple LLM-generated tool-and-action plans without evolutionary refinement”, we already included this experiment as an ablation in section 6.2 of our paper.
>
> **Question 2: Justification for action representation.**
> We adopted discrete waypoints to help effectively (and with limited compute) evaluate our two main hypotheses:
>
> (1) Thoughtful tool design can solve robotic manipulation problems even with very simple actions
>
> (2) Modern VLMs can guide tool design within the right framework, i.e., VLMgineer.
>
> Discrete kinematic configuration waypoints are merely one convenient choice of action space: they permit fast, feed-forward VLM synthesis of compact action plan sketches for each synthesized tool, enabling us to quickly evaluate a large number of candidate designs with minimal compute. In practice, this allows VLMgineer to generate and evaluate both tools and associated action plans for new tasks in under 30 minutes with our limited compute resources (maybe say what our compute looked like?), entirely without task-specific engineering or tuning
>
> **Question 3: Separating tool-quality vs. action-quality effects.**
> In VLMgineer, tool and action are explicitly co-optimized as a single individual, so attributing failures to “tool quality” versus “action quality” in isolation is inherently ill-defined. The evolutionary loop selects and mutates pairs of (tool, waypoint trajectory) based solely on the joint behavioral outcome, and our contribution is precisely this coupled adaptation. While one could design additional ablations that fix one component and vary the other, our main claim concerns the performance of the co-designed system, rather than decomposing responsibility between its two tightly coupled parts.
>
> **Question 1: Quantitative benefit of joint tool–action sampling.**
>
> As we pointed out in our response to Q3, tools are difficult to evaluate without actions, unless the human provides manual guidance or evaluation metrics. Since our main purpose is to automate the tool generation process, adding a human-in-the-loop step would not result in a fair comparison.
>
> **Question 5: Evolutionary search vs. simpler iterative refinement.**
> The performance of evolutionary search substantially benefits from the diversity of parallel sampling in each iteration. In contrast, we have empirically observed that simple iterative refinement lacks sample diversity, and consequently, the VLM would often be stuck in a local minimum. Furthermore, since only one sample is generated per API call, and only one sample is evaluated at once, iterative refinement is also expensive and slow. We will show these conclusions by experiment.
>
> **Question 4: Clarity in Figure 6\.**
> Thanks for pointing this out. The more transparent light blue and orange represent the maximum reward, while the opaque bars represent the average reward. This will be revised and improved in our next PDF.
>
> **Weakness 5: Ambiguity in Table 1 metrics.**
> We appreciate the reviewer's feedback on the clarity of our metrics. The values reported in Table 1 represent **normalized task rewards** (continuous progress metrics scaled between 0 and 1), as defined in Section 6 ('Evaluation Metrics') and detailed for each task in Appendix A.2.
>
> However, we agree that this distinction is critical, as normalized rewards can be easily mistaken for binary success rates. We will update the caption and headers of Table 1 to explicitly state 'Mean Normalized Reward (0–1)' to ensure immediate clarity for readers.
>
> **Weakness 1**: **Citation formatting reduces readability.**
> Thank you for pointing this out. We will change the citation format to a more readable version.

---

### Official Review · Reviewer_UWYg · 2025-11-01

**Soundness:** 2
**Presentation:** 3
**Contribution:** 2
**Rating:** 4
**Confidence:** 5

**Summary:**

This paper presents VLMGINEER, a system that uses a VLM to invent new tools for robots. It uses an "evolutionary search" process where the VLM generates, tests, and improves tool designs and the robot actions needed to use them. The main idea is to create tools that make the robot's task feasible or easier. The paper also introduces ROBOTOOLBENCH, a new 12-task simulation benchmark for this problem. Results show this method performs better than a standard robot gripper and tool designs specified by human experts. The authors also successfully transferred three of the simulated designs to a real robot.

**Strengths:**

- The paper studies the challenging problem of automatic tool-action co-design. This is a significant and creative step beyond typical VLM-based planning, which generally assumes a set of pre-existing tools. This task requires a high degree of physical common-sense reasoning, and the paper demonstrates that VLMs can provide useful priors for this.

- The proposed method is well-explained. It effectively uses the VLM's semantic and physical knowledge to guide the search, successfully automating the process that would typically require significant hand-engineering.

- The ROBOTOOLBENCH test suite is a valuable contribution. Its 12 tasks are well-design and could be useful for future research.

**Weaknesses:**

- The framework's reliance on thousands of simulation-based evaluations for its evolutionary loop (e.g., 8000 samples for the ablation ) makes the full design process impractical for direct real-world deployment. The paper demonstrates sim-to-real transfer on only 3 of the 12 tasks. More critically, these transfers rely on open-loop execution of discrete waypoints, which is known to be brittle and highly sensitive to any real-world physics or calibration error. This remains a major gap between the simulated success and a real-world "robotic toolsmith."
- The set of baselines is a significant weakness. The primary comparisons are against "Human Prompts" and existing "RLBench Tools". While outperforming the "Human Prompts" baseline is interesting, it mainly proves that evolutionary search is superior to a single-shot (human + VLM) guess. The related work (Section 2) extensively discusses prior computational co-design methods, including those using RL , differentiable simulation , and model-based optimization. Liu et al. (2023) is even cited as inspiration for a task. However, none of these state-of-the-art computational methods are included as quantitative baselines. Without them, it is impossible to situate VLMGINEER's performance relative to the actual prior art in computational tool design.
- The "Human Prompts" baseline, which is central to the paper's claim of outperforming human specifications, is not a rigorous study. The appendix (A.1.1) reveals this was a "case study" with only three participants: one "LLM expert," one "robotics expert," and one "layperson". This sample size is far too small to draw any generalizable or statistically meaningful conclusions.

**Questions:**

- The sim-to-real transfer was open-loop. How robust are these policies to minor perturbations in the real-world setup (e.g., a 1cm shift in object position)?
- The action representation is an $N \times 7$ array of waypoints. How is $N$ (the number of waypoints) determined? Is it a fixed hyperparameter per task, or does the VLM decide on the number of waypoints? This seems like a critical parameter that would heavily influence the performance.
- For example, how often does the VLM produce an invalid URDF, a non-manipulable tool, or a design that violates a constraint? How does the pipeline handle such errors?
- Number of trials for each human subject, what information is presented, what training the participants received, and how the participants were recruited?

---

> ### Author Response · Authors · 2025-11-26
> **Rebuttal by Authors**
>
> We sincerely thank the reviewer for their thoughtful and constructive feedback.
>
> **Weakness 1 \- “Gap between the simulated success and a real-world 'robotic toolsmith’."**
>
> First, we believe our approach is best interpreted in the context of sim-to-real policy transfer, which is a well-established paradigm. We require an approximate digital twin to serve as a ground truth environment for the evolutionary search, allowing us to safely evaluate and refine tool-policy pairs before physical deployment. This reliance on a simulation environment is in line with other works in sim-to-real and Large Model-guided evolutionary search, citing works such as Eureka\[1\] or RialTo\[2\], which similarly leverage raw environment code to ground the model's reasoning in the underlying physics of the task. However, rather than using simulation to synthesize policies alone, we demonstrate the possibility of transferring co-designed tools and policies.
>
> Second, while 8,000 samples seem plenty, VLMgineer is already considerably more sample and time-efficient than previous methods. Prior work (e.g. \[3\] Appendix B.3) with similar tasks requires more time and compute: 24 hours with 1e7 samples for each task in PyBullet. By contrast, VLMgineer only requires 30 minutes with \~6,000 samples (see L1042), where each sample corresponds to 8 action waypoints. Also, VLMginner is without human-in-the-loop steps such as task-specific templates, prompts, or examples, whereas these aforementioned prior works require humans to specify optimization parameters and templates.
>
> **Weakness 2: “Set of baselines is a significant weakness”**
>
> Thank you. We would have loved to compare to more baselines, but we are not aware of any that would be appropriate comparisons. As mentioned at line 126 of the paper, the studies in the Related Work section all involve substantial manual parameter tuning or predefined parametric tool designs, which are fundamentally different from our fully automated approach. Hence, direct apples-to-apples comparisons would be challenging. In fact, our method could serve as a strong prior for these related works. In the revised version (will upload in the next few days), we will explicitly state the challenges associated with direct comparison to parametric design methods and clearly articulate the advantages of our fully automated and parameter-free approach.
>
> **Weakness 2 \+ Weakness 3: “It mainly proves that evolutionary search is superior to a single-shot (human \+ VLM) guess” and “Small Sample Size for Human Case Study”**
>
> We acknowledge that our sample size is too small to support strong, generalizable claims about "super-human" performance. In the final revision, we will temper this language to accurately reflect that this was a preliminary case study rather than a rigorous human-subject evaluation. We are also currently expanding this baseline with a larger and more diverse participant pool to provide more robust statistical evidence.
>
> However, we maintain that the current experiments offer significant scientific value as a demonstration of **non-triviality** of results on this challenging autonomous co-design problem. The fact that our fully automated method consistently outperforms designs guided by an LLM expert, a robotics expert, and a layperson suggests that ***the task is not easily solved by a human simply "asking" a VLM***. The core contribution of VLMgineer is not necessarily that it is "smarter" than all humans, but that it successfully removes the human from the loop entirely while achieving performance that is not achievable by trivial human efforts.
>
> **(Continued below...)**

---

> ### Author Response · Authors · 2025-11-26
> **Rebuttal by Authors**
>
> **Question 1: Robustness to “minor perturbations in the real-world setup”**
>
> Please first refer to our response to Weakness 1
>
> Additionally, we emphasize that our sim‑to‑real evaluations were conducted under carefully controlled conditions, with object positions matched to the simulation, to demonstrate the initial feasibility of the synthesized tool and action sequences. Robustness to real‑world perturbations was not our main objective; rather, we sought to demonstrate a proof of concept for transferring synthesized tools and action plans to real robots. Moreover, there exists a substantial body of prior work on sim‑to‑real policy transfer, which we could readily build upon to handle robustness and generalization in future extensions.
>
> **Question 2: Waypoint numbers**
>
> The number of waypoints the VLM outputs is not a parameter that we decide; it is fully determined by the VLM. We can revise and make this detail clearer.
>
> **Question 3: How does our pipeline handle invalid URDFs?**
>
> Our experience indicates that the VLM outputs invalid URDFs not very often due to careful prompt engineering within system instructions. When invalid URDFs are produced, our pipeline quickly identifies and discards these designs. Specifically, our system manages URDF syntactical errors by assigning a fitness score of 0 to invalid designs, effectively discarding them from subsequent evolutionary iterations. Moreover, straightforward checks for issues like self-collision and constraint violations can be implemented to further ensure the validity of designs, and such validation methods are simple to incorporate into the pipeline.
>
> **Question 4: Details on Human Participant Studies**
>
> The details of our human‑prompted design experiment are provided in Appendix A.1.1. For each human subject, we collected a prompt for each task and ran the pipeline for five trials under the same task conditions. Participants were recruited through a volunteer call to graduate‑school students. Prior to the experiments, each participant received a briefing that described the overall procedure, the task goal, and the scope of allowed manipulations. We did not provide additional “training”, instead, we showed each participant the simulation environment (including object scale, workspace layout, and target goal) so they could reasonably anticipate the scale and physical constraints of the task. We also allowed Q\&A before the experiments. This was meant to ensure that their prompts are credible within the simulated physical context.
>
> \[1\] Ma, Yecheng Jason, et al. "Eureka: Human-Level Reward Design via Coding Large Language Models." The Twelfth International Conference on Learning Representations.
>
> \[2\] Torne, Marcel, et al. "Reconciling reality through simulation: A real-to-sim-to-real approach for robust manipulation." arXiv preprint arXiv:2403.03949 (2024).
>
> \[3\] Liu, Ziang, et al. "Learning to Design and Use Tools for Robotic Manipulation." Conference on Robot Learning. PMLR, 2023\.

---

### Official Review · Reviewer_Cg95 · 2025-11-01

**Soundness:** 2
**Presentation:** 3
**Contribution:** 2
**Rating:** 4
**Confidence:** 4

**Summary:**

The paper proposes a fully automatic framework, VLMgineer, that designs both tools and corresponding actions from scratch by drawing on the generative capabilities of Vision–Language Models (VLMs) together with evolutionary search. The system jointly discovers tool geometries and manipulation strategies to solve tool-use tasks, and it is evaluated on a custom benchmark (RoboToolBench) of everyday manipulation scenarios where creative tool design is considered necessary. While the idea of co-optimizing tool design and use is conceptually interesting, the practical benefits and generalizability of the approach remain unclear. The experimental setup lacks strong baselines, and the model relies too heavily on task-specific and highly detailed inputs, which limits its applicability and generalization to broader scenarios.

**Strengths:**

- Using evolutionary strategies to generate tools, rather than a pure VLM solution, is a strength that introduces an iterative refinement process, helping to generate diverse and more functional designs.

- RoboToolBench provides a set of distinct environments tailored for evaluating the performance of tool-use strategies.

- The method yields more effective designs and improved performance for the given task compared to human-prompted designs.

**Weaknesses:**

- The tools are designed as permanent end-effector attachments (or new grippers), rather than as distinct, portable tools that would be grasped by the robot. This design choice fundamentally limits the system’s generalizability and applicability to real-world scenarios where a robot must use a variety of pre-existing, non-integrated tools.

- The overall evaluation lacks sufficient diversity and strength in its baselines. The comparison to the default two-fingered Franka Panda gripper is weak, as these tasks are explicitly designed to be unsolvable without custom tools. While the work includes a comparison to RLBench tools on four tasks, it omits comparisons against recent, more sophisticated computational co-design or tool-learning baselines (e.g., one of the studies reported in the Related Work section), which would provide a more robust validation of VLMgineer's effectiveness. The comparison against a small-sample group of three human designers is also subjective.

- The VLM requires low-level and environment-specific information, including raw environment code, task description, and explicit system instructions as input (Figure 2). This reliance on low-level, internal data severely limits the method's generalizability and reusability, as a truly general-purpose system should ideally operate with high-level, natural inputs like an environment image and a simple task goal.

- The authors claim to have empirically observed their action sampling strategy outperforms other forms of feedback (line 212); however, no quantitative analysis or representative baseline is provided in the paper to substantiate this critical statement.

- The authors claim that optimizing both the tool’s design and its manipulation strategy is a more practical solution than using already existing tools in the environment for a general-purpose robot (line 111). However, this assertion is questionable regarding real-world practicality, scalability, and generalization. Continuously generating and 3D printing new tools for each task is costly and time-consuming (line 437), limiting its applicability. An approach focused on learning when to use existing tools and when to generate/form new ones  would contribute more significantly to generalizable robotics and open-world applications.

- For clarity and reproducibility, the paper should include an example of the generated action sequences (the 7 waypoints) and a discussion of their corresponding usage and limitations.

- It would be better to present the algorithm in the main paper rather than in the Appendix.

**Questions:**

In addition to the issues presented in the weakness section:

- The paper compares VLMgineer’s co-designed tool and action against the original, pre-existing tools from RLBench. Since the RLBench tools have established, fixed designs and VLMGINEER can co-optimize both tool and action, is this a fair comparison? Specifically, the robot cannot improve the pre-established tools in the RLBench environment setup, which limits the optimization space compared to VLMGINEER's full co-design approach. Could the authors clarify the intent behind this specific comparison?

- The VLMgineer framework takes "System Instructions" as input, alongside the raw environment code and task description. For clarity and to better assess the generalizability of the VLM prompt, what exactly do these system instructions entail? Given that the model already leverages a Vision-Language Model, wouldn't it be more generalizable if the inputs were ideally limited to high-level, natural inputs like the environment image and a simple task goal to maximize generalizability and reusability?

---

> ### Author Response · Authors · 2025-11-26
> **Rebuttal by Authors**
>
> We thank the reviewer for the detailed and insightful feedback.
>
> **Weakness 2 \- “Compare to computational co-design or tool learning baselines from the related work section?”**
>
> As mentioned at line 126 of the paper, the studies in the Related Work section all involve substantial manual parameter tuning or predefined parametric tool designs, which are fundamentally different from our fully automated approach. Hence, direct apples-to-apples comparisons would be challenging. In fact, our method could serve as a strong prior for these related works. In the revised version (will upload in the next few days), we will explicitly state the challenges associated with direct comparison to parametric design methods and clearly articulate the advantages of our fully automated and parameter-free approach.
>
> The comparison to human designs serves to showcase the advantages of automation and the potential of our method used as a prior, but it is not intended as a definitive baseline. We will clarify this in the revision.
>
> **Weakness 3 & Question 2 \- “The VLM requires low-level and environment-specific information, including raw environment code, task description, and explicit system instructions as input” and “Clarification on system instructions”**
>
> We believe that our approach is best interpreted in the context of sim-to-real policy transfer, which is a well-established paradigm. Therefore, similar to other work, Eureka \[1\] or RialTo \[2\] in the sim2real context, we also rely on the existence of a simulation environment and, naturally, by extension, the environment code. However, while these and many other works have attempted to build digital twins of the real world or training policies in simulation that are transferable to the real world, we instead investigate a much less studied aspect: automating the design of tools and how to wield them.  Our focus is on designing a fully autonomous framework for this co-design problem, minimizing any task-specific manual design.  We leave integrating well-established modules, such as digital twin constructions and sim2real policy training, to future work. We will add a note to the paper to avoid this confusion.
>
> Also note that the "System Instructions" are just the meta-prompts that define the VLM's role, valid output formats (e.g., URDF specifications), and design constraints. They do not contain task-specific solutions but rather ensure the model generates executable and physically valid candidates. Please see Appendix A.4 for the full prompt. While we agree that visual-only input is a desirable long-term goal, current foundational models benefit significantly from accompanying visual information with the precise physical context provided by code to reason effectively about contact dynamics and geometry.
>
> \[1\] Ma, Yecheng Jason, et al. "Eureka: Human-Level Reward Design via Coding Large Language Models." The Twelfth International Conference on Learning Representations.
>
> \[2\] Torne, Marcel, et al. "Reconciling reality through simulation: A real-to-sim-to-real approach for robust manipulation." arXiv preprint arXiv:2403.03949 (2024).
>
> **Weakness 1 \- “Permanent end-effector attachments rather than portable tools that would be grasped by the robot.”**
>
> The main purpose of VLMgineer is to demonstrate the plausibility of VLM-guided physical tool and action co-design, establishing the fundamental first step for a tall ladder of automating commercial-grade physical design deployable in real-world factories. We fully agree that designing tools as graspable objects rather than permanent attachments would make our method more general and practical; we hope to achieve this as we climb this ladder in future work. In fact, we can easily integrate sampling today’s advanced grasp foundation models \[1\]\[2\]  for grasp generation into our action sampling framework. Furthermore, end-effector attachments also enable VLMs to be creative about incorporating the gripper’s degree of freedom in tool design: the *SnatchCookie* task in our RoboToolBench already explicitly involves controlling the gripper, demonstrating our method's capability in this regard.
>
> \[1\] Murali, Adithyavairavan, et al. "Graspgen: A diffusion-based framework for 6-dof grasping with on-generator training." arXiv preprint arXiv:2507.13097 (2025).
>
> \[2\] Yuan, Wentao, et al. "M2T2: Multi-Task Masked Transformer for Object-centric Pick and Place." Conference on Robot Learning. PMLR, 2023\.
>
>
> **(Continued below...)**

---

> ### Author Response · Authors · 2025-11-26
> **Rebuttal by Authors (Continued)**
>
> **Weakness 5 \- Practicality and scalability concerns regarding generating new tools in the real world**
>
> We appreciate the reviewer’s concerns regarding real-world deployments. We agree and will revise our wording to clarify our intent. We meant that it may not always be feasible to assume that a convenient tool already exists in the environment for every new task. Instead, the ability to design and fashion new tools suited to a task is much more general. We also acknowledge that integrating existing or previously designed tools from past tasks is beneficial, and future integration with methods capable of selecting between existing and newly generated tools could indeed enhance practicality and scalability.
>
> **Weakness 4 \- No quantitative analysis of the action sampling strategy**
>
> We will include these quantitative results explicitly in the revision to substantiate our claims.
>
> Our system intentionally employed selection via reward as the primary evolution guide in our framework because we empirically observed it to be the most effective at evolving into better designs. This choice was made after empirically testing several forms of feedback, none of which provided meaningful improvement, or sometimes hurt performance.
>
> For example, we explored feeding the result image directly back into the VLM prompt at each iteration. Surprisingly, we found this often harmed performance, reducing success rates. Careful inspection indicated that the VLM frequently misinterpreted visual inputs, struggling to correctly identify task success/failure or interpret subtle interaction results from raw visual feedback alone, leading to bad samples surviving through the next evolution. We also experimented with more structured feedback, such as using object-state and bounding boxes to represent the task progress. Although this explicitly encoded rich information, in practice, the VLM struggled with grounding fine-grained numerical or geometric data to infer task progress. Therefore, it misjudged the design's performance, which in turn hurt the next evolution as a bad tool could still survive.
>
> On the other hand, scalar rewards naturally condense the most relevant task information into a concise and easily interpretable form for VLM. As such, VLM is guaranteed to receive the most successful designs from the last iteration as in-context examples for the next iteration of evolution.
>
> **Weakness 6 & Weakness 7 \- Action sequences example and formatting**
>
> We agree that providing explicit examples of generated action sequences will enhance clarity. We will include representative examples of action sequences. We will include the VLMgineer algorithm in the main text in a revised version of our PDF.
>
> **Question 1 \- “Could the authors clarify the intent behind the RLBench comparison?”**
>
> We will add an acknowledgment that RLBench tools, although manually designed, may not have been optimized explicitly to maximize task success rates. Nevertheless, these tools typically represent sensible and commonly used designs for their respective tasks, as shown in Fig. 5, providing a meaningful and practical benchmark. We will clarify this intent explicitly in our revision.

---

### Author Response · Authors · 2025-12-03
**General Rebuttal Summary (Part 1)**

Dear ACs, SACs, PCs, and Reviewers,

We sincerely thank you for the time and effort in reviewing our submission. Our work presents VLMgineer, a fully automatic framework that designs tools and actions without human-in-the-loop specifications by harnessing the creativity of Vision–Language Models together with evolutionary search. We demonstrated strong empirical results across a diverse tool manipulation benchmark (RoboToolBench) and further demonstrated that VLMgineer’s automatically designed tools and action policies transfer to real-world task execution on a physical robot.

During the discussion phase, we made comprehensive efforts to address all major concerns raised by each reviewer. Specifically, we clarified methodological differences with prior works, addressed concerns about baselines, provided detailed explanations on design choices, and enhanced clarity. We have submitted a revised manuscript with changes highlighted in blue.

**Major common concerns:**

**Baseline Comparisons:**
Reviewers (```Cg95```, ```UWYg```, ```gsFY```) pointed out **potential missing baseline comparisons**. Simply put, all prior **methods require human-in-the-loop specifications of tool parameters and templates, and thus direct comparisons to them** are fundamentally challenging since VLMgineer is **fully automatic**. Regarding the human prompt baseline, we explicitly acknowledged its illustrative nature, and we committed to clarifying its role more transparently in the manuscript. Additionally, we discussed concurrent work (RobotSmith), clearly articulating that VLM is used only as a modular component to initialize conventional optimization in RobotSmith, whereas we involve VLM throughout our pipeline to unleash its full creative potential.

**Real-World Practical Considerations:**
Reviewers (```Cg95```, ```gsFY```, ```UWYg```, ```5kyU```) raised points regarding real-world scalability and practical considerations, particularly our use of open-loop action execution and concerns about privileged information requirements. We respond by emphasizing that our work is best interpreted in the context of **sim-to-real policy transfer**, which is a **well-established paradigm that permits the use of privileged information**. While we acknowledge future work should explore a closed-loop policy that suits transfer better, we want to point out that this is not the focus of our contribution, which remains a novel VLM-based evolution design framework.

**More Ablations:**
Additionally, several reviewers (```Cg95```, ```gsFY```) requested further justification for our evolutionary framework's design, specifically questioning the choice of feedback modality and the necessity of population-based evolution over simpler iterative refinement. In response, we performed **additional experiments** on the effect of **incorporating image feedback** as a part of evolution, as well as on **repeatedly refining a single sample**. The former yielded 5.4% lower reward than our method, while the latter yielded 49.1 % lower performance, definitively justifying our current design choices. Please refer to our Appendix section A.11 and A.12 in our revised PDF for the exact experimental setup and results.

**(Continued below)**

---

> ### Author Response · Authors · 2025-12-03
> **General Rebuttal Summary (Part 2)**
>
> **Individual Reviewer Concerns:**
>
> **Reviewer ```Cg95```:**
>
> We clarified that direct comparisons with existing computational co-design methods are challenging due to the manual parameter tuning or predefined parametric tool designs these methods require, while our approach is fully automated **(Weakness 2\)**. We also justified our use of low-level and environment-specific information as standard practice aligned with sim-to-real research, with clarification on “system prompt” being task-agnostic **(Weakness 3 & Question 2\)**. Additional explanation is provided for the practical implications of our permanent tool attachment design and future integration with graspable tools **(Weakness 1\)**. We provide quantitative results justifying our choice of scalar rewards as feedback, leading to better performance than other forms of feedback, such as images **(Weakness 4\)**. Additionally, we clarified our intent behind the RLBench comparison, acknowledging that VLMgineer could be used to design performance-optimized tools that outperform everyday tools deemed appropriate for tasks. **(Question 1\)**.
>
> **Reviewer ```UWYg```:**
>
> In our response to reviewer ```UWYg```, we clarified that our sim-to-real evaluations were carefully controlled demonstrations intended to showcase the feasibility of transferring co-designed tools and action plans to physical robots, rather than comprehensive robustness tests **(Weakness 1 & Question 1\)**. We explained our method's efficiency relative to previous approaches and highlighted distinctions: we are the first fully automatic approach without the need for human-in-the-loop specifications, which makes direct baseline comparisons challenging **(Weakness 2\)**. We acknowledged the limited human baseline size, clarifying its illustrative intent and noting ongoing efforts to expand for stronger validation **(Weakness 2 & Weakness 3\)**. Additionally, we stated that waypoint numbers are autonomously determined by the VLM and added an example in Appendix A.6 **(Question 2\)**, explained our handling of invalid URDF designs **(Question 3\)**, and provided clear details on participant and experimental setup for human-prompted design studies **(Question 4\)**.
>
> **Reviewer ```gsFY```:**
>
> In our response to reviewer ```gsFY```, we addressed the concern regarding concurrent work RobotSmith **(Weakness 2\)**, emphasizing that, unlike RobotSmith’s VLM usage followed by conventional optimization, our approach uses the VLM as a closed‑loop co‑designer, and it is engaged throughout our evolution pipeline, allowing flexible modifications such as part editing and reconfiguration. Regarding novelty claims **(Weakness 3\)**, we clarified that our contribution in evolutionary search surpasses single-shot iterative prompting, confirmed by additional experiments in Appendix A.11. **(Question 5\)**. We also noted our focus on low-level tool-action co-design rather than long-horizon tasks **(Weakness 4\)**, acknowledged manufacturability constraints as future work **(Weakness 6\)**, explained waypoint action representations **(Question 2\)**, clarified joint evaluations on tool and action **(Question 1 & Question 3\)**, and improved citation readability **(Weakness 1\)**.
>
> **Reviewer ```5kyU```:**
>
> In response to reviewer ```5kyU```, we clarified that our method aligns with established sim-to-real paradigms, emphasizing our unique focus on autonomous tool-action co-design and validating its feasibility **(Weakness 1\)**. We explained that VLMgineer still needs to reason about object geometry, contact forces, collision avoidance, etc., despite the provided environment code **(Question 1\)**. We highlighted key differences from prior LLM-based morphology studies **(Weakness 2\)**. Lastly, regarding using human/RLBench tools as priors, we hypothesized limited diversity from prior but committed to exploring this in the future **(Question 2\)**
>
> We hope we have carefully considered and addressed each reviewer’s concerns. We believe the revised manuscript (changes highlighted in blue) clearly reflects these improvements and demonstrates the novelty of our contributions. We kindly ask the AC to take these clarifications and improvements into consideration during evaluation.
>
> Sincerely,
> The Authors.

---

### Meta-Review · Area_Chair_kDGz · 2026-01-04

**Summary:**

This paper introduces a fully autonomous framework for co-designing robot end-effector tool geometry and corresponding action sequences using vision–language models (VLMs). Experiments on 12 RLBench-style manipulation tasks demonstrate that the proposed method consistently discovers task-specific tools and action plans that outperform both human-designed tools and the default RLBench tools.

Reviewers raised concerns about the practicality of the approach, particularly the assumption of access to full environment code and the reliance on expensive VLM queries in the evolution iterations. Additional concerns were expressed regarding the strength of the baselines. In the rebuttal, the authors addressed these points by providing further ablation studies that justify key design choices and by clarifying that most prior approaches inherently require human-in-the-loop design or task-specific parameterization, whereas this work focuses on a fully autonomous and complementary paradigm for tool evolution.

While an iterative human baseline allowing human designers to refine their tools over multiple rounds could further strengthen the empirical comparison, this does not detract from the core contribution. Overall, the paper presents a compelling and novel direction for autonomous robot tool design, supported by strong experimental evidence, and therefore I recommend acceptance.

**Reviewer Concerns:**

The reviewers acknowledge the novelty and effectiveness of the proposed method in simulation, but raise significant concerns about its practical applicability, given the strong assumptions of full task knowledge and accurate simulation. They also question the fairness of the comparisons with human-designed and default RLBench tool baselines, and several reviewers note that additional ablation studies are needed to better justify the design choices.

**Reviewer Scores:**

During rebuttal, the authors provided detailed clarifications and well-reasoned responses to each of the raised concerns, along with additional ablation results that strengthen the empirical evidence. As a result, it is likely that the initially weak rejections could turn positive given more time for discussion.

---

### Decision · Program_Chairs · 2026-01-26

Accept (Poster)